# OFFLINE META-REINFORCEMENT LEARNING WITH ADVANTAGE WEIGHTING

## ABSTRACT

This paper introduces the offline meta-reinforcement learning (offline meta-RL) problem setting and proposes an algorithm that performs well in this setting. Offline meta-RL is analogous to the widely successful supervised learning strategy of pre-training a model on a large batch of fixed, pre-collected data (possibly from various tasks) and fine-tuning the model to a new task with relatively little data. That is, in offline meta-RL, we meta-train on fixed, pre-collected data from several tasks and adapt to a new task with a very small amount (less than 5 trajectories) of data from the new task. By nature of being offline, algorithms for offline meta-RL can utilize the largest possible pool of training data available and eliminate potentially unsafe or costly data collection during meta-training. This setting inherits the challenges of offline RL, but it differs significantly because offline RL does not generally consider a) transfer to new tasks or b) limited data from the test task, both of which we face in offline meta-RL. Targeting the offline meta-RL setting, we propose Meta-Actor Critic with Advantage Weighting (MACAW). MACAW is an optimization-based meta-learning algorithm that uses simple, supervised regression objectives for both the inner and outer loop of meta-training. On offline variants of common meta-RL benchmarks, we empirically find that this approach enables fully offline meta-reinforcement learning and achieves notable gains over prior methods.

## 1 INTRODUCTION

Meta-reinforcement learning (meta-RL) has emerged as a promising strategy for tackling the high sample complexity of reinforcement learning algorithms, when the goal is to ultimately learn many tasks. Meta-RL algorithms exploit shared structure among tasks during meta-training, amortizing the cost of learning across tasks and enabling rapid adaptation to new tasks during meta-testing from only a small amount of experience. Yet unlike in supervised learning, where large amounts of pre-collected data can be pooled from many sources to train a single model, existing meta-RL algorithms assume the ability to collect millions of environment interactions *online* during meta-training. Developing *offline* meta-RL methods would enable such methods, in principle, to leverage existing data from any source, making them easier to scale to real-world problems where large amounts of data might be necessary to generalize broadly. To this end, we propose the offline meta-RL problem setting and a corresponding algorithm that uses only offline (or batch) experience from a set of training tasks to enable efficient transfer to new tasks without any further interaction with *either* the training or testing environments. See Figure 1 for a comparison of offline meta-RL and standard meta-RL.

Because the offline setting does not allow additional data collection during training, it highlights the desirability of a *consistent* meta-RL algorithm. A meta-RL algorithm is consistent if, given enough diverse data on the test task, adaptation can find a good policy for the task regardless of the training task distribution. Such an algorithm would provide a) rapid adaptation to new tasks from the same distribution as the train tasks while b) allowing for improvement even for out of distribution test tasks. However, designing a consistent meta-RL algorithm in the offline setting is difficult: the consistency requirement suggests we might aim to extend the model-agnostic meta-learning (MAML) algorithm (Finn et al., 2017a), since it directly corresponds to fine-tuning at meta-test time. However, existing MAML approaches use online policy gradients, and only value-based approaches have proven effective in the offline setting. Yet combining MAML with value-based RL subroutines is not straightforward: the higher-order optimization in MAML-like methods demands stable and efficient gradient-descent updates, while TD backups are both somewhat unstable and require a large number of steps to propagate reward information across long time horizons.

To address these challenges, one might combine MAML with a supervised, bootstrap-free RL subroutine, such as advantage-weighted regression (AWR) (Peters and Schaal, 2007; Peng et al., 2019), for both for the inner and outer loop of a gradient-based meta-learning algorithm, yielding a 'MAML+AWR' algorithm. However, as we will discuss in Section 4 and find empirically in Section 5, naïvely combining MAML and AWR in this way does not provide satisfactory performance because the AWR policy update is not sufficiently expressive. Motivated by prior work that studies the expressive power of MAML (Finn and Levine, 2018), we increase the expressive power of the meta-learner by introducing a carefully chosen policy update in the inner loop. We theoretically prove that this change increases the richness of the policy's update and find empirically that this policy update dramatically improves adaptation performance and stability in some settings. We further observe that standard feedforward neural network architectures used in reinforcement learning are not well-suited to optimization-based meta-learning and suggest an alternative that proves critical for good performance across four different environments. We call the resulting meta-RL algorithm and architecture Meta-Actor Critic with Advantage Weighting, or MACAW.

Our main contributions are the offline meta-RL problem setting itself and MACAW, an offline meta-reinforcement learning algorithm that possesses three key properties: sample efficiency, offline meta-training, and consistency at meta-test time. To our knowledge, MACAW is the first algorithm to successfully combine gradient-based meta-learning and off-policy value-based RL. Our evaluations include experiments on offline variants of standard continuous control meta-RL benchmarks as well as settings specifically designed to test the robustness of an offline meta-learner when training tasks are scarce. In all of these settings, MACAW significantly outperforms fully offline variants state-of-the-art off-policy RL and meta-RL baselines.

## 2 PRELIMINARIES

In reinforcement learning, an agent interacts with a Markov Decision Process (MDP) to maximize its cumulative reward. An MDP is a tuple $(\mathcal{S}, \mathcal{A}, T, r)$ consisting of a state space $\mathcal{S}$, an action space $\mathcal{A}$, stochastic transition dynamics $T : \mathcal{S} \times \mathcal{A} \times \mathcal{S} \rightarrow [0, 1]$, and a reward function $r$. At each time step, the agent receives reward $r_t = r(s_t, a_t, s_{t+1})$. The agent's objective is to maximize the expected return (i.e. discounted sum of rewards) $\mathcal{R} = \sum_t \gamma^t r_t$, where $\gamma \in [0, 1]$ is a discount factor. To extend this setting to meta-RL, we consider tasks drawn from a distribution $\mathcal{T}_i \sim p(\mathcal{T})$, where each task $\mathcal{T}_i = (\mathcal{S}, \mathcal{A}, p_i, r_i)$ represents a different MDP. Both the dynamics and reward function may vary across tasks. The tasks are generally assumed to exhibit some (unknown) shared structure. During meta-training, the agent is presented with tasks sampled from $p(\mathcal{T})$; at meta-test time, an agent's objective is to rapidly find a high-performing policy for a (potentially unseen) task $\mathcal{T}' \sim p(\mathcal{T})$. That is, with only a small amount of experience on $\mathcal{T}'$, the agent should find a policy that achieves high expected return on that task. During meta-training, the agent meta-learns parameters or update rules that enables such rapid adaptation at test-time.

**Model-agnostic meta-learning** One class of algorithms for addressing the meta-RL problem (as well as meta-supervised learning) are variants of the MAML algorithm (Finn et al., 2017a), which involves a bi-level optimization that aims to achieve fast adaptation via a few gradient updates. Specifically, MAML optimizes a set of initial policy parameters $\theta$ such that a few gradient-descent steps from $\theta$ leads to policy parameters that achieve good task performance. At each meta-training step, the inner loop adapts $\theta$ to a task $\mathcal{T}$ by computing $\theta' = \theta - \alpha \nabla_\theta \mathcal{L}_{\mathcal{T}}(\theta)$, where $\mathcal{L}$ is the loss function for task $\mathcal{T}$ and $\alpha$ is the step size (in general, $\theta'$ might be computed from multiple gradient steps, rather than just one as is written here). The outer loop updates the initial parameters as $\theta \leftarrow \theta - \beta \nabla_\theta \mathcal{L}'_{\mathcal{T}}(\theta')$, where $\mathcal{L}'_{\mathcal{T}}$ is a loss function for task $\mathcal{T}$, which may or may not be the same as the inner-loop loss function $\mathcal{L}_{\mathcal{T}}$, and $\beta$ is the step size. MAML has been previously instantiated with policy gradient updates in the inner and outer loops (Finn et al., 2017a; Rothfuss et al., 2018), which can only be applied to on-policy meta-RL settings; we address this shortcoming in this work.

**Advantage-weighted regression.** To develop an offline meta-RL algorithm, we build upon advantage-weighted regression (AWR) (Peng et al., 2019), a simple offline RL method. The AWR policy objective is given by

$$\mathcal{L}^{\text{AWR}}(\vartheta, \varphi, B) = \mathbb{E}_{\mathbf{s}, \mathbf{a} \sim B} \left[ -\log \pi_\vartheta(\mathbf{a}|\mathbf{s}) \exp \left( \frac{1}{T} \left( \mathcal{R}_B(\mathbf{s}, \mathbf{a}) - V_\varphi(\mathbf{s}) \right) \right) \right], \qquad (1)$$

where $B = \{\mathbf{s}_j, \mathbf{a}_j, \mathbf{s}'_j, r_j\}$ can be an arbitrary dataset of transition tuples sampled from some behavior policy, and $\mathcal{R}_B(\mathbf{s}, \mathbf{a})$ is the return recorded in the dataset for performing action $\mathbf{a}$ in state

Figure 1: Comparing the standard meta-RL setting (left), which includes on-policy and off-policy meta-RL, with offline meta-RL (right). In standard meta-RL, new interactions are sampled from the environment during both meta-training and meta-testing, potentially storing experiences in a replay buffer (off-policy meta-RL). In *offline* meta-RL, a batch of data is provided for each training task $\mathcal{T}_i$. This data could be the result of prior skills learned, demonstrations, or other means of data collection. The meta-learner uses these static buffers of data for meta-training and can then learn a new test task when given a small buffer of data for that task.

$\mathbf{s}$, $V_\varphi(\mathbf{s})$ is the learned value function for the behavior policy evaluated at state $\mathbf{s}$, and $T > 0$ is a temperature parameter. The term $\mathcal{R}_B(\mathbf{s}, \mathbf{a}) - V_\varphi(\mathbf{s})$ represents the advantage of a particular action. The objective can be interpreted as a weighted regression problem, where actions that lead to higher advantages are assigned larger weights. The value function parameters $\varphi$ are typically trained using simple regression onto Monte Carlo returns, and the policy parameters $\vartheta$ are trained using $\mathcal{L}^{\mathrm{AWR}}$. Next, we discuss the offline meta-RL problem and some of the challenges it poses.

## 3  THE OFFLINE META-RL PROBLEM

In the offline meta-RL problem setting, we aim to leverage offline multi-task experience to enable fast adaptation to new downstream tasks. Each task $\mathcal{T}_i$ is drawn from a task distribution $p(\mathcal{T})$. In the offline setting, the meta-training algorithm is not permitted to directly interact with the meta-training tasks $\mathcal{T}_i$, but instead is provided with a fixed dataset of transition tuples $B_i = \{s_{i,j}, a_{i,j}, s'_{i,j}, r_{i,j}\}$ for each task. Each $B_i$ is populated with trajectories sampled from a corresponding behavior policy $\mu_i$. Each $\mu_i$ might be an expert policy, sub-optimal demonstrations, other RL agents, or some mixture thereof. Regardless of the behavior policies $\mu_i$, the objective of offline meta-RL is to maximize return after adaptation on the test tasks. However, depending on the quality of the behavior policies, the maximum attainable return may vary. We observe such a phenomenon in a offline data quality ablation experiment in Section 5.

Sampling data from a fixed dataset at both meta-training and meta-testing time, rather than from the learned policy itself, distinguishes offline meta-RL from the standard meta-RL setting. This constraint is significant, because most algorithms for meta-RL require a large amount of on-policy experience from the environment during meta-training; these algorithms are generally unable to fully make use of data collected by external sources. During meta-testing, a (generally unseen) test task $\mathcal{T}_{\mathrm{test}}$ is drawn from $p(\mathcal{T})$, and the meta-trained agent is presented with a new batch of experience $D$ sampled from a distribution $B_{\mathrm{test}}$. The agent's objective is to use this batch of data to find the highest-performing policy for the test task. We consider the case where only $B_i$ is fixed during meta-training and $B_{\mathrm{test}}$ corresponds to sampling online trajectories to be the *offline meta-RL problem*. The case where both $B_i$ and $B_{\mathrm{test}}$ are fixed data buffers is called the *fully offline meta-RL problem*, which is especially applicable in situations when allowing online exploration might be difficult or dangerous. In the fully offline case, we might also consider the setting where we perform additional online rollouts with our adapted policy and fine-tune with this online data after the initial offline adaptation step. We call this the fully offline meta-RL problem *with online fine-tuning*. The experiments performed in this paper mostly correspond to the fully offline setting. In Appendix C.2 we also conduct an experiment in the setting of fully offline meta-RL with online fine-tuning.

Prior meta-RL methods require interaction with the MDP for each of the meta-training tasks (Finn et al., 2017a), and though some prior methods build on off-policy RL algorithms (Rakelly et al., 2019), these algorithms are known to perform poorly in the fully offline setting (Levine et al., 2020). Both of the offline meta-RL settings described above inherit the distributional difficulties of offline RL, which means that addressing this problem setting requires a new type of meta-RL method that is capable of meta-training from offline data.

## 4  MACAW: META ACTOR-CRITIC WITH ADVANTAGE WEIGHTING

---

**Algorithm 1** MACAW Meta-Training

1: **Input:** Tasks $\{\mathcal{T}_i\}$, offline buffers $\{D_i\}$
2: **Hyperparameters**: learning rates $\alpha_1, \alpha_2, \eta_1,$ $\eta_2$, training iterations $n$, temperature $T$
3: Randomly initialize meta-parameters $\theta, \phi$
4: **for** $n$ steps **do**
5:     **for** task $\mathcal{T}_i \in \{\mathcal{T}_i\}$ **do**
6:         Sample disjoint batches $D_i^{\text{tr}}, D_i^{\text{ts}} \sim D_i$
7:         $\phi_i' \leftarrow \phi - \eta_1 \nabla_\phi \mathcal{L}_V(\phi, D_i^{\text{tr}})$
8:         $\theta_i' \leftarrow \theta - \alpha_1 \nabla_\theta \mathcal{L}_\pi(\theta, \phi_i', D_i^{\text{tr}})$
9:     $\phi \leftarrow \phi - \eta_2 \sum_i \left[ \nabla_\phi \mathcal{L}_V(\phi_i', D_i^{\text{ts}}) \right]$
10:    $\theta \leftarrow \theta - \alpha_2 \sum_i \left[ \nabla_\theta \mathcal{L}^{\text{AWR}}(\theta_i', \phi_i', D_i^{\text{ts}}) \right]$

---

**Algorithm 2** MACAW Meta-Testing

1: **Input:** Test task $\mathcal{T}_j$, offline experience $D$, meta-policy $\pi_\theta$, meta-value function $V_\phi$
2: **Hyperparameters**: learning rates $\alpha_1, \eta$, adaptation iterations $n$, temperature $T$
3: Initialize $\theta_0 \leftarrow \theta, \phi_0 \leftarrow \phi$.
4: **for** $n$ steps **do**
5:     $\phi_{t+1} \leftarrow \phi_t - \eta_1 \nabla_{\phi_t} \mathcal{L}_V(\phi_t, D)$
6:     $\theta_{t+1} \leftarrow \theta_t - \alpha_1 \nabla_{\theta_t} \mathcal{L}_\pi(\theta_t, \phi_{t+1}, D)$

---

In addition to satisfying the demands of the offline setting, an ideal method for offline meta-RL should not be limited to the distribution of tasks observed at training time. This is especially important in the offline meta-RL setting, in which the sampling of the training data is out of the control of the agent. In other words, it is critical that an offline meta-RL algorithm be consistent, in the sense that given enough, sufficiently diverse adaptation data at meta-test time, the algorithm can find a good solution to that task, regardless of the meta-training tasks.

To address the numerous challenges posed by offline meta-RL, we propose meta actor-critic with advantage weighting (MACAW). MACAW is an offline meta-RL algorithm that learns initializations $\phi$ and $\theta$ for a value function $V_\phi$ and policy $\pi_\theta$, respectively, that can rapidly adapt to a new task seen at meta-test time via gradient descent. Both the value function and the policy objectives correspond to simple regression losses in both the inner and outer loop, leading to a stable and consistent inner-loop adaptation process and outer-loop meta-training signal. While these objectives build upon AWR, we show that the naive application of an AWR update in the inner loop leads to unsatisfactory performance, motivating the enriched policy update that we describe in Section 4.1. In Sections 4.2 and 4.3, we detail the full meta-training procedure and an important architectural component of the policy and value networks.

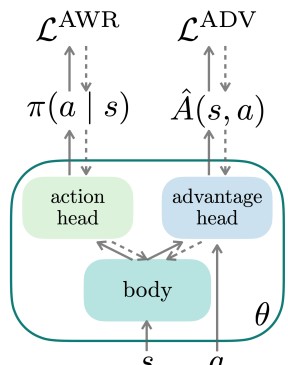

Figure 2: MACAW policy architecture. Solid lines show the forward pass; dashed lines show gradient flow during the backward pass *during adaptation only*; the advantage head is not used in the outer loop policy update.

### 4.1 INNER-LOOP MACAW PROCEDURE

The adaptation process for MACAW consists of a value function update followed by a policy update and can be found in lines 6-8 in Algorithm 1. Optimization-based meta-learning methods typically rely on truncated optimization for the adaptation process (Finn et al., 2017a), to satisfy both computational and memory constraints (Wu et al., 2018; Rajeswaran et al., 2019), and MACAW also uses a truncated optimization. However, value-based algorithms that use bootstrapping, such as Q-learning, can require many iterations for values to propagate. Therefore, we use a bootstrap-free update for the value function that simply performs supervised regression onto Monte-Carlo returns.

Given a batch of training data $D_i^{\text{tr}}$ collected for $\mathcal{T}_i$, MACAW adapts the value function by taking one or a few gradient steps on the following supervised objective:

$$\phi_i' \leftarrow \phi - \eta_1 \nabla_\phi \mathcal{L}_V(\phi, D_i^{\text{tr}}), \qquad \text{where} \qquad \mathcal{L}_V(\phi, D) \triangleq \mathbb{E}_{\mathbf{s}, \mathbf{a} \sim D} \left[ (V_\phi(\mathbf{s}) - \mathcal{R}_D(\mathbf{s}, \mathbf{a}))^2 \right] \quad (2)$$

and where $\mathcal{R}_D(\mathbf{s}, \mathbf{a})$ is the Monte Carlo return from the state $\mathbf{s}$ taking action $\mathbf{a}$ observed in $D$.

After adapting the value function, we proceed to updating the policy. The AWR algorithm updates its policy by performing supervised regression onto actions weighted by the estimated advantage, where the advantage is given by the return minus the value: $\mathcal{R}_D(\mathbf{s}, \mathbf{a}) - V_{\phi_i'}(\mathbf{s})$. While it is tempting to use this same update rule here, we observe that this update does not provide the meta-learner with sufficient expressive power to be a universal update procedure for the policy, using universality in the sense used by Finn and Levine (2018). For MAML-based methods to approximate any learning procedure, the inner gradient must not discard information needed to infer the task (Finn and Levine,

2018). The gradient of the AWR objective does not contain full information of both the regression weight and the regression target. That is, one cannot recover both the advantage weight and the action from the gradient. We formalize this problem in Theorem 1 in Appendix A. To address this issue and make our meta-learner sufficiently expressive, the MACAW policy update performs both advantage-weighted regression onto actions as well as an additional regression onto action advantages. This enriched policy update is only used during adaptation, and the predicted advantage is used only to enrich the inner loop policy update during meta-training; during meta-test, this predicted advantage is discarded. We prove the universality of the enriched policy update in Theorem 2 in Appendix A. We observe empirically the practical impact of the universality property with an ablation study presented in Figure 4 (left).

To make predictions for both the AWR loss and advantage regression, our policy architecture has two output heads corresponding to the predicted action given the state, $\pi_\theta(\cdot|\mathbf{s})$, and the predicted advantage given both state and action $A_\theta(\mathbf{s}, \mathbf{a})$. This architecture is shown in Figure 2. Policy adaptation then proceeds as follows:

$$\theta_i' \leftarrow \theta - \alpha_1 \nabla_\theta \mathcal{L}_\pi(\theta, \phi_i', D_i^{\text{tr}}), \qquad \text{where} \qquad \mathcal{L}_\pi = \mathcal{L}^{\text{AWR}} + \lambda \mathcal{L}^{\text{ADV}}. \tag{3}$$

In our policy update, we show only one gradient step for conciseness of notation, but it can be easily extended to multiple gradient steps. The AWR loss is given in Equation 1, and the advantage regression loss is given by:

$$\mathcal{L}^{\text{ADV}}(\theta, \phi_i', D) \triangleq \mathop{\mathbb{E}}_{\mathbf{s}, \mathbf{a} \sim D} \left[ (\hat{A}(\mathbf{s}, \mathbf{a}) - (\mathcal{R}_D(\mathbf{s}, \mathbf{a}) - V_{\phi_i'}(\mathbf{s})))^2 \right] \tag{4}$$

Adapting with $\mathcal{L}_\pi$ rather than $\mathcal{L}^{\text{AWR}}$ addresses the expressiveness problems noted earlier. This adaptation process is done both in the inner loop of meta-training and during meta-test time, as outlined in Algorithm 2. MACAW is *consistent* at meta-test time because it executes a well-defined RL fine-tuning subroutine based on AWR during adaptation. Next, we describe the meta-training procedure for learning the meta-parameters $\theta$ and $\phi$, the initializations of the policy and value function, respectively.

## 4.2 OUTER-LOOP MACAW PROCEDURE

To enable rapid adaptation at meta-test time, we meta-train a set of initial parameters for both the value function and policy to optimize the AWR losses $\mathcal{L}_V$ and $\mathcal{L}^{\text{AWR}}$, respectively, after adaptation (L9-10 in Algorithm 1). We sample a batch of data $D_i^{\text{ts}}$ for the outer loop update that is disjoint from the adaptation data $D_i^{\text{tr}}$ in order to promote few-shot generalization rather than memorization of the adaptation data. The meta-learning procedure for the value function follows MAML, using the supervised Monte Carlo objective:

$$\min_\phi \mathbb{E}_{\mathcal{T}_i} \left[ \mathcal{L}_V(\phi_i', D_i^{\text{ts}}) \right] \;=\; \min_\phi \mathbb{E}_{\mathcal{T}_i} \left[ \mathcal{L}_V(\phi - \eta_1 \nabla_\phi \mathcal{L}_V(\phi, D_i^{\text{tr}}), D_i^{\text{ts}}) \right]. \tag{5}$$

where $\mathcal{L}_V$ is defined in Equation 2. This objective optimizes for a set of initial value function parameters such that one or a few inner gradient steps lead to an accurate value estimator.

Unlike the inner loop, we optimize the initial policy parameters in the outer loop with a standard advantage-weighted regression objective, since expressiveness concerns only pertain to the inner loop where only a small number of gradient steps are taken. Hence, the meta-objective for our initial policy parameters is as follows:

$$\min_\theta \mathbb{E}_{\mathcal{T}_i} \left[ \mathcal{L}^{\text{AWR}}(\theta_i', \phi_i', D_i^{\text{ts}}) \right] \;=\; \min_\theta \mathbb{E}_{\mathcal{T}_i} \left[ \mathcal{L}^{\text{AWR}}(\theta - \alpha_1 \nabla_\theta \mathcal{L}_\pi(\theta, \phi_i', D_i^{\text{tr}}), \phi_i', D_i^{\text{ts}}) \right], \tag{6}$$

where $\mathcal{L}_\pi$ is defined in Equation 3 and $\mathcal{L}^{\text{AWR}}$ is defined in Equation 1. Note we use the adapted value function for policy adaptation. The complete MACAW algorithm is summarized in Algorithm 1.

## 4.3 MACAW ARCHITECTURE

MACAW's enriched policy update (Equation 3) is motivated by the desire to make inner loop policy updates more expressive. In addition to augmenting the objective, we can also take an architectural approach to increasing gradient expressiveness. Recall that for an MLP, a single step of gradient descent can only make a rank-1 update to each weight matrix. Finn and Levine (2018) show that this implies that MLPs must be impractically deep for MAML to be able to produce *any* learning

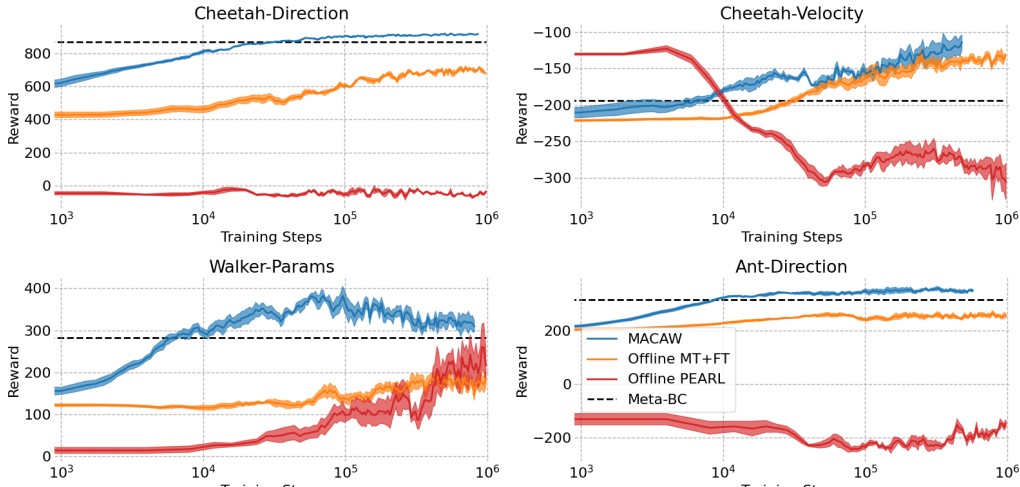

Figure 3: Comparing MACAW with (i) an offline variant of PEARL (Rakelly et al., 2019), a state-of-the-art off-policy meta-RL method, (ii) an offline multi-task training + fine tuning method based on AWR (Peng et al., 2019), and (iii) a meta-behavior cloning baseline. Shaded regions show one standard error of the mean reward of four seeds. MACAW is the only algorithm to consistently outperform the imitation learning baseline, and also learns with the fewest number of training steps in every environment (note the log x axis).

procedure. However, we can shortcut this rank-1 limitation with a relatively simple change to the layers of an MLP, which we call a *weight transform* layer. This layer maps a latent code into the weight matrix and bias, which are then used to compute the layer's output just as in a typical fully-connected layer. This 'layer-wise linear hypernetwork' (Ha et al., 2016) doesn't change the class of functions computable by the layer on its input, but it increases the expressivity of MAML's gradient. Because we update the latent code by gradient descent (which is mapped back into a new weight matrix and bias in the forward pass) we can, in theory, acquire weight matrix updates of rank up to the dimensionality of the latent code. We use this strategy for all of the weights in both the value function network and the policy network. This architecture is similar to latent embedding optimization (LEO) (Rusu et al., 2019), but the choice of using simple linear mapping functions allows us to apply weight transform layers to the entire network while still providing more expressive gradients. For a more detailed explanation of this strategy, see Appendix B. Our experiments find that this layer significantly improves learning speed and stability.

## 5 EXPERIMENTS

The primary goal of our empirical evaluations is to test whether we can acquire priors from offline multi-task data that facilitate rapid transfer to new tasks. Our evaluation compares MACAW with three sensible approaches to this problem: a meta-imitation learning, multi-task offline RL with fine-tuning, and an offline variant of the state-of-the-art off-policy meta-RL method, PEARL (Rakelly et al., 2019). Further, we analyze a) the importance of MACAW's enriched policy update (Equation 3) in various data quality regimes; b) the effect of the proposed weight transformation; and c) how each method's performance is affected when the sampling of the task space during training is very sparse. The first two settings highlight the differences between MACAW and the naïve combination of MAML and AWR; the third setting represents a realistic setting where fewer tasks are available during meta-training. **See Appendix C for additional experiments** a) ablating the weight transform layer b) investigating the performance of MACAW and PEARL when online fine-tuning is available and c) a richer task distribution.

For our experiments, we construct offline variants of the widely-used simulated continuous control benchmark problems introduced by Finn et al. (2017a); Rothfuss et al. (2018), including the half-cheetah with varying directions and varying velocities, the walker with varying physical parameters, and the ant with varying directions. If not noted otherwise, the offline data for each experiment is generated from the replay buffer of a RL agent trained from scratch. This reflects a practical scenario where an agent has previously learned a set of tasks via RL, stored its experiences, and now would like to quickly learn a related task. Data collection information is available in Appendix D.

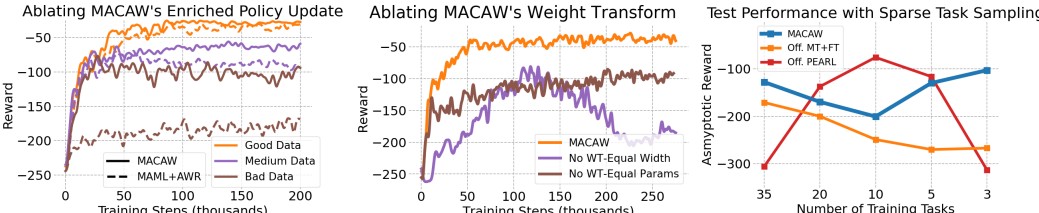

Figure 4: **Left**: Ablating MACAW's enriched policy update when varying the quality of the **inner loop** adaptation data. Solid lines correspond to MACAW, dashed lines correspond to MACAW without the auxiliary policy loss (equivalently, MAML+AWR with weight transforms). Both perform similarly with good quality adaptation data (orange), but the policy adaptation step without the auxiliary loss begins to fail as adaptation data is increasingly sub-optimal (blue and red). Bad, medium, and good data correspond to the first, middle, and last 500 trajectories from the lifetime replay buffer of the behavior policy for each task; see Appendix D for learning curves of the individual offline policies. **Center**: Ablating MACAW's weight transform layer in the same experimental setting as the cheetah-velocity experiment in Figure 3. Without the additional expressiveness, learning is much slower and less stable. **Right**: Train task sparsity split performance of MACAW, Offline PEARL, and Offline MT+fine tune. Each curve corresponds to the performance of a method as the number of tasks available for training is varied. MACAW shows the most consistent performance when different numbers of tasks are used, performing well even when only **three tasks** are used for training.

**Can we learn to adapt to new tasks quickly from purely offline data?**   Our first evaluation compares three approaches to the offline meta-RL problem setting, testing their ability to leverage the offline task datasets in order to quickly adapt to a new task. Specifically, we compare MACAW with i) offline PEARL (Rakelly et al., 2019), ii) multi-task AWR (Peng et al., 2019), which uses 20 steps of Adam (Kingma and Ba, 2015) to adapt to a new task at meta-test time (Offline MT+FT) and iii) a meta-behavior cloning baseline. We choose PEARL and AWR because they achieve state-of-the-art performance in off-policy meta-RL and offline RL, respectively, and are readily adaptable to the offline meta-RL problem. As in Rakelly et al. (2019), for each experiment, we sample a finite set of training tasks and held out test tasks upfront and keep these fixed throughout training. Figure 3 shows the results. We find that MACAW is the only algorithm to consistently outperform the meta-behavior cloning baseline. Multi-task AWR + fine-tuning makes meaningful progress on the simpler cheetah problems, but it is unable to adapt well on the more challenging walker and ant problems. Offline PEARL shows initial progress on cheetah-velocity and walker-params, but struggles to make steady progress on any of the problems. We attribute PEARL's failure to Q-function extrapolation error, a problem known to affect many off-policy RL algorithms (Fujimoto et al., 2019), as well as generally unstable offline bootstrapping. MACAW's and AWR's value function is bootstrap-free and their policy updates maximize a weighted maximum likelihood objective during training, which biases the policy toward safer actions (Peng et al., 2019), implicitly avoiding problems caused by extrapolation error. In contrast to Offline PEARL and multi-task AWR, MACAW trains efficiently and relatively stably on all problems, providing an effective approach to learning representations from multi-task offline data that can be effectively adapted to new tasks at meta-test time.

**How does MACAW's performance differ from MAML+AWR?**   MACAW has two key features distinguishing it from MAML+AWR: the enriched policy loss and weight transform layers. Here, we use the Cheetah-Velocity setting to test the effects of both of these changes. We first ablate the enriched policy loss used in MACAW's inner loop update. This experiment compares MACAW and MAML+AWR+weight transform layers, which optimize Equation 3 and Equation 1 in the policy inner-loop, respectively. To identify when policy update expressiveness is most crucial, we repeat this ablation study three times, meta-training and meta-testing with various qualities of inner-loop data, using good outer loop data for all experiments. Figure 4 (left) shows the results. MAML+AWR performs well when the offline adaptation data comes from a near-optimal policy, which is essentially a one-shot imitation setting (orange); however, when the offline adaptation data comes from a policy pre-convergence, the difference between MACAW and MAML+AWR becomes significant (blue and red). This result supports the intuition that policy update expressiveness is of greater importance when the adaptation data is more random, because in this case the adaptation data includes a weaker signal from which to infer the task (e.g. the task cannot be inferred by simply looking at the states visited). Because an agent is unable to collect further experience from the environment during offline adaptation, it is effectively at the mercy of the quality of the behavior policy that produced the data. An important property of a meta-RL algorithm is thus its robustness to sub-optimal behavior policies, a property that MACAW exhibits. Next, we ablate the weight transform layers, comparing

MAML+AWR+enriched policy update with MACAW. Figure 4 (center) suggests that the weight transform layers significantly improve both learning speed and stability. The No WT-Equal Width variant removes the weight transform from each fully-connected layer, replacing it with a regular fully-connected layer of equal width in the forward pass. The No WT-Equal Params variant replaces each of MACAW's weight transform layers with a regular fully-connected layer of greater width, to keep the total number of learnable parameters in the network roughly constant. In either case, we find that MACAW provides a significant improvement in learning speed, as well as stability when compared to the Equal Width variant. Figure 5 in the appendix shows that this result is consistent across problems.

**How do algorithms perform with varying numbers of meta-training tasks?** Generally, we prefer an offline meta-RL algorithm that can generalize to new tasks when presented with only a small number of meta-training tasks sampled from $p(\mathcal{T})$. In this section, we conduct an experiment to evaluate the extent to which various algorithms rely on dense sampling of the space of tasks during training in order to generalize well. We compare the test performance of MACAW, offline PEARL, and offline multi-task AWR + fine-tuning as we hold out an increasingly large percentage of the Cheetah-Velocity task space. The results are presented in Figure 4 (right). Surprisingly, Offline PEARL completely fails to learn both when training tasks are plentiful and when they are scarce, but learns relatively effectively in the middle regime (5-20 tasks). In our experiments, we often observe instability in Offline PEARL's task inference and value function networks when training on too many offline tasks. On the other hand, with too few tasks, the task inference network simply learns a degenerate solution, providing no useful information for the value functions or policy to identify the task. The multi-task learning + fine-tuning baseline exhibits a steadier degradation in performance as training tasks are removed, likely owing to its bootstrap-free learning procedure. Similarly to Offline PEARL, it is not able to learn a useful prior for fine-tuning when only presented with 3 tasks for training. However, MACAW finds a solution of reasonable quality for any sampling of the task space, even for very dense or very sparse samplings of the training tasks. In practice, this property is desirable, because it allows the same algorithm to scale to very large offline datasets while still producing useful adaptation behaviors for small datasets. Ultimately, MACAW effectively exploits the available data when meta-training tasks are plentiful and shows by far the greatest robustness when tasks are scarce, which we attribute to its SGD-based adaptation procedure during both meta-training and meta-testing.

## 6 RELATED WORK

Meta-learning algorithms enable efficient learning of new tasks by learning elements of the learning process itself (Schmidhuber, 1987; Bengio et al., 1992; Thrun and Pratt, 1998; Finn, 2018). We specifically consider the problem of meta-reinforcement learning. Prior methods for meta-RL can generally be categorized into two groups. Contextual meta-RL methods condition a neural network on experience using a recurrent network (Wang et al., 2016; Duan et al., 2016; Fakoor et al., 2020), a recursive network (Mishra et al., 2017), or a stochastic inference network (Rakelly et al., 2019; Zintgraf et al., 2020; Humplik et al., 2019; Sæmundsson et al., 2018). Optimization-based meta-RL methods embed an optimization procedure such as gradient descent into the meta-level optimization (Finn et al., 2017a; Nagabandi et al., 2019; Rothfuss et al., 2018; Zintgraf et al., 2019; Gupta et al., 2018; Mendonca et al., 2019; Yang et al., 2019), potentially using a learned loss function (Houthooft et al., 2018; Bechtle et al., 2019; Kirsch et al., 2020b;a). In prior works, the former class of approaches tend to reach higher asymptotic performance, while the latter class is typically more robust to out-of-distribution tasks, since the meta-test procedure corresponds to a well-formed optimization. Concurrent work by Dorfman and Tamar (2020) investigates the offline meta-RL setting, directly applying an existing meta-RL algorithm, VariBAD (Zintgraf et al., 2020), to the offline setting. The proposed method further assumes knowledge of the reward function for each task to relabel rewards and share data across tasks with shared dynamics. MACAW does not rely on this knowledge nor the assumption that some tasks share dynamics, but this technique could be readily combined with MACAW when these assumptions do hold.

Unlike these prior works, we aim to develop an optimization-based meta-RL algorithm that can both learn from entirely offline data and produces a monotonic learning procedure. Only a handful of previous model-free meta-RL methods leverage off-policy data at all (Rakelly et al., 2019; Mendonca et al., 2019), and none have considered the fully offline setting. Guided meta-policy search (Mendonca et al., 2019) is optimization-based, but is not applicable to the batch setting as it partially relies on policy gradients. This only leaves PEARL (Rakelly et al., 2019) and its relatives (Fakoor et al., 2020),

which correspond to a contextual meta-learning approach that is sensitive to the meta-training task distribution without fine-tuning (Fakoor et al., 2020) at test time. We also compare to PEARL, and find that, as expected, it performs worse than in the off-policy setting, since the fully offline setting is substantially more challenging than the off-policy setting that it was designed for.

The proposed algorithm builds on the idea of batch off-policy or offline reinforcement learning (Fujimoto et al., 2019; Kumar et al., 2019b; Wu et al., 2019; Levine et al., 2020; Agarwal et al., 2020), extending the problem setting to the meta-learning setting. There are a number of recent works that have demonstrated successful results with offline reinforcement learning and deep neural networks (Fujimoto et al., 2019; Jaques et al., 2019; Kumar et al., 2019a; Wu et al., 2019; Peng et al., 2019; Agarwal et al., 2020). We specifically choose to build upon the advantage-weighted regression (AWR) algorithm (Peng et al., 2019). We find that AWR performs well without requiring dynamic programming, instead using Monte Carlo estimation to infer the value function. This property is appealing, as it is difficult to combine truncated optimization-based meta-learners such as MAML (Finn et al., 2017a) with TD learning, which requires a larger number of gradient steps to effectively back-up values.

## 7 CONCLUSION

In this work, we formulated the problem of offline meta-reinforcement learning and presented MACAW, a practical algorithm that achieves good performance on various continuous control tasks compared with other state-of-the-art meta-RL algorithms. We motivated the design of MACAW by the desire to build an offline meta-RL algorithm that is both sample-efficient (using value-based RL subroutines) and consistent (running a full-fledged RL algorithm at test time). We hope that this work serves as the basis for future research in offline meta-RL, enabling more sample-efficient learning algorithms to make better use of purely observational data from previous tasks and adapt to new tasks more quickly.

We consider fully offline meta-training and meta-testing with and without online fine-tuning, showing that MACAW is effective both when collecting online data is totally infeasible as well as when some online data collection is possible at meta-test time. However, an interesting direction for future work is to consider how we might enable online adaptation from purely offline meta-training while preserving the consistency property of MACAW. This would require an offline strategy for learning to explore, a problem that has largely been considered in on-policy settings in the past (Gupta et al., 2018; Zintgraf et al., 2020) but also recently in offline settings (Dorfman and Tamar, 2020).

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

# Appendix

## A   MACAW AUXILIARY LOSS AND UPDATE EXPRESSIVENESS

Finn and Levine (2018) lay out conditions under which the MAML update procedure is universal, in the sense that it can approximate any function $f(\mathbf{x}, \mathbf{y}, \mathbf{x}^*)$ arbitrarily well (given enough capacity), where $\mathbf{x}$ and $\mathbf{y}$ are the support set inputs and labels, respectively, and $\mathbf{x}^*$ is the test input. Universality in this sense is an attractive property because it implies that the update is expressive enough to approximate any update procedure; a method that does *not* possess the universality property might be limited in its asymptotic post-adaptation performance because it cannot express (or closely approximate) the true optimal update procedure. In order for the MAML update procedure to be universal, several requirements of the network architecture, hyperparameters, and loss function must be satisfied. Most of these are not method-specific in that they stipulate minimum network depth, activation functions, and non-zero learning rate for any neural network. However, the condition placed on the loss function require more careful treatment. The requirement is described in Definition 1.

**Definition 1.** *A loss function is 'universal' if the gradient of the loss with respect to the prediction(s) is an invertible function of the label(s) used to compute the loss.*

We note that Definition 1 is a necessary but not sufficient condition for an update procedure to be universal (see other conditions above and Finn and Levine (2018)). For the AWR loss function (copied below from Equation 1 with minor changes), the labels are the ground truth action $\mathbf{a}$ and the corresponding advantage $\mathcal{R}(\mathbf{s}, \mathbf{a}) - V_{\phi_i'}(\mathbf{s})$.

$$\mathcal{L}^{\text{AWR}}(\mathbf{s}, \mathbf{a}, \theta, \phi_i') = -\log \pi_\theta(\mathbf{a}|\mathbf{s}) \exp\left( \frac{1}{T} \left( \mathcal{R}(\mathbf{s}, \mathbf{a}) - V_{\phi_i'}(\mathbf{s}) \right) \right) \tag{7}$$

For simplicity and without loss of generality (see Finn and Levine (2018), Sections 4 & 5), we will consider the loss for only a single sample, rather than averaged over a batch.

In the remainder of this section, we first state in Theorem 1 that the standard AWR policy loss function does not satisfy the condition for universality described in Definition 1. The proof is by a simple counterexample. Next, we state in Theorem 2 that the MACAW auxiliary loss does satisfy the universality condition, enabling a universal update procedure given the other generic universality conditions are satisfied (note that the MACAW value function loss satisfies the condition in Definition 1 because it uses L2 regression Finn and Levine (2018)).

### A.1   NON-UNIVERSALITY OF STANDARD AWR POLICY LOSS FUNCTION

Intuitively, the AWR gradient does not satisfy the invertibility condition because it does not distinguish between a small error in the predicted action that has a large corresponding advantage weight and a large error in the predicted action (in the same direction) that has a small corresponding advantage weight. The following theorem formalizes this statement.

**Theorem 1.** *The AWR loss function $\mathcal{L}^{AWR}$ is not universal according to Definition 1.*

The proof is by counterexample; we will show that there exist different sets of labels $\{\mathbf{a}_1, A_1(\mathbf{s}, \mathbf{a}_1)\}$ and $\{\mathbf{a}_2, A_2(\mathbf{s}, \mathbf{a}_1)\}$ that produce the same gradient for some output of the model. First, rewriting Equation 7 with $A(\mathbf{s}, \mathbf{a}) = \left( \mathcal{R}(\mathbf{s}, \mathbf{a}) - V_{\phi_i'}(\mathbf{s}) \right)$, we have

$$\mathcal{L}^{\text{AWR}}(\mathbf{s}, \mathbf{a}, \theta) = -\log \pi_\theta(\mathbf{a}|\mathbf{s}) \exp\left( \frac{A(\mathbf{s}, \mathbf{a})}{T} \right)$$

Because our policy is parameterized as a Gaussian with fixed diagonal covariance $\sigma^2 I$, we can again rewrite this loss as

$$\mathcal{L}^{\text{AWR}}(\mathbf{s}, \mathbf{a}, \hat{\mathbf{a}}_\mu) = \left( \log \frac{1}{(2\pi\sigma^2)^{\frac{k}{2}}} + \frac{||\mathbf{a} - \hat{\mathbf{a}}_\mu||^2}{2\sigma^2} \right) \exp\left( \frac{A(\mathbf{s}, \mathbf{a})}{T} \right) \tag{8}$$

where $\hat{\mathbf{a}}_\mu$ is the mean of the Gaussian output by the policy and $k = \dim(\mathbf{a})$. For the purpose of the simplicity of the counterexample, we assume the policy output $\hat{\mathbf{a}}_\mu$ is $\mathbf{0}$. The gradient of this loss with respect to the policy output is

$$\nabla_{\hat{\mathbf{a}}_\mu} \mathcal{L}^{\text{AWR}}(\mathbf{s}, \mathbf{a}, \mathbf{0}) = -\frac{1}{\sigma^2} \exp\left( \frac{A(\mathbf{s}, \mathbf{a})}{T} \right) \mathbf{a}$$

To demonstrate that the gradient operator applied to this loss function is not invertible, we pick two distinct label values and show that they give the same gradient. We pick $\mathbf{a}_1 = [1, ..., 1]^T$, $A_1(\mathbf{s}, \mathbf{a}_1) = T$ and $\mathbf{a}_2 = [0.1, ..., 0.1]^T$, $A_2(\mathbf{s}, \mathbf{a}_2) = \log(10)T$. Inserting these values into Equation A.1, this gives gradients $g_1 = \frac{-e}{\sigma^2}[1, ..., 1]^T$ and $g_2 = \frac{-10e}{\sigma^2}[0.1, ..., 0.1]^T = \frac{-e}{\sigma^2}[1, ..., 1]^T = g_1$. Thus the gradient of the standard AWR loss does not possess sufficient information to recover the labels uniquely and using this loss for policy adaptation does not produce a universal policy update procedure. Next, we show how the auxiliary loss used in MACAW alleviates this problem.

## A.2   UNIVERSALITY OF THE MACAW POLICY ADAPTATION LOSS FUNCTION

In this section, we show that by adding an additional term to the AWR loss function, we acquire a loss that satisfies the condition stated in Definition 1, which we state in Theorem 2. Intuitively, the additional loss term allows the gradient to distinguish between the cases that were problematic for the AWR loss (large action error and small advantage weight vs small action error and large advantage weight).

**Theorem 2.** *The MACAW policy loss function $\mathcal{L}_\pi$ is universal according to Definition 1.*

The MACAW policy adaptation loss (given in Equation 3) is the sum of the AWR loss and an auxiliary advantage regression loss (the following is adapted from Equation 8):

$$\mathcal{L}_\pi(\mathbf{s}, \mathbf{a}, \hat{\mathbf{a}}_\mu, \hat{A}) = \left( \log \frac{1}{(2\pi\sigma^2)^{\frac{k}{2}}} + \frac{||\mathbf{a} - \hat{\mathbf{a}}_\mu||^2}{2\sigma^2} \right) \exp\left( \frac{A(\mathbf{s}, \mathbf{a})}{T} \right) + \lambda(A(\mathbf{s}, \mathbf{a}) - \hat{A})^2$$

where $\hat{A}$ is the predicted advantage output from the policy advantage head and $\lambda$ is the advantage regression coefficient. The gradient of this loss with respect to the predicted advantage $\hat{A}$ is

$$g_{\text{ADV}} = \nabla_{\hat{A}} \mathcal{L}_\pi(\mathbf{s}, \mathbf{a}, \hat{\mathbf{a}}_\mu, \hat{A}) = 2\lambda(\hat{A} - A(\mathbf{s}, \mathbf{a})) \tag{9}$$

and the gradient of the loss with respect to $\hat{\mathbf{a}}_\mu$ is

$$\mathbf{g}_{\text{AWR}} = \nabla_{\hat{\mathbf{a}}_\mu} \mathcal{L}_\pi(\mathbf{s}, \mathbf{a}, \hat{\mathbf{a}}_\mu, \hat{A}) = \frac{1}{\sigma^2} \exp\left( \frac{A(\mathbf{s}, \mathbf{a})}{T} \right) (\hat{\mathbf{a}}_\mu - \mathbf{a}) \tag{10}$$

We write the combined gradient as $\mathbf{g} = \begin{bmatrix} g_{\text{ADV}} \\ \mathbf{g}_{\text{AWR}} \end{bmatrix}$. In order to provide a universal update procedure, we must be able to recover both the action label $\mathbf{a}$ and the advantage label $A(\mathbf{s}, \mathbf{a})$ from $\mathbf{g}$. First, because $g_{\text{ADV}}$ is an invertible function of $A(\mathbf{s}, \mathbf{a})$, we can directly extract the advantage label by re-arranging Equation 9:

$$A(\mathbf{s}, \mathbf{a}) = \frac{g_{\text{ADV}} - 2\lambda\hat{A}}{-2\lambda}$$

Similarly, $\mathbf{g}_{\text{AWR}}$ is an invertible function of $\mathbf{a}$, so we can then extract the action label by re-arranging Equation 10:

$$\mathbf{a} = \frac{\mathbf{g}_{\text{AWR}} - \frac{1}{\sigma^2} \exp\left( \frac{A(\mathbf{s}, \mathbf{a})}{T} \right) \hat{\mathbf{a}}_\mu}{-\frac{1}{\sigma^2} \exp\left( \frac{A(\mathbf{s}, \mathbf{a})}{T} \right)} \tag{11}$$

Because we can compute $A(\mathbf{s}, \mathbf{a})$ from $g_{\text{ADV}}$, there are no unknowns in the RHS of Equation 11 and we can compute $\mathbf{a}$ (here, $\sigma$, $\lambda$, and $T$ are known constants); it is thus the additional information provided by $g_{\text{ADV}}$ that resolves the ambiguity that is problematic for the standard AWR policy loss gradient. We have now shown that both the action label and advantage label used in the MACAW policy adaptation loss are recoverable from its gradient, implying that the update procedure is universal under the conditions given by Finn and Levine (2018), which concludes the proof.

## B   WEIGHT TRANSFORM LAYERS

Here, we describe in detail the 'weight transformation' layer that augments the expressiveness of the MAML update in MACAW. First, we start with the observation in past work (Finn et al., 2017b) that adding a 'bias transformation' to each layer improves the expressiveness of the MAML update.

To understand the bias transform, we compare with a typical fully-connected layer, which has the forward pass

$$\mathbf{y} = \sigma\left(W\mathbf{x} + \mathbf{b}\right)$$

where $\mathbf{x}$ is the previous layer's activations, $\mathbf{b}$ is the bias vector, $W$ is the weight matrix, and $\mathbf{y}$ is this layer's activations. For a bias transformation layer, the forward pass is

$$\mathbf{y} = \sigma\left(W\mathbf{x} + W^b\mathbf{z}\right)$$

where $\mathbf{z}$ and $W^b$ are learnable parameters of the bias transformation. During adaptation, either only the vector $\mathbf{z}$ or both the vector $\mathbf{z}$ and the bias matrix $W^b$ are adapted. The vector $W^b\mathbf{z}$ has the same dimensionality as the bias in the previous equation. This formulation does not increase the expressive power of the forward pass of the layer, but it does allow for a more expressive update of the 'bias vector' $W^b\mathbf{z}$ (in the case of dim($\mathbf{z}$) = dim($\mathbf{b}$) and $W^b = I$, we recover the standard fully-connected layer).

For a weight transformation layer (used in MACAW), we extend the idea of computing the bias from a latent vector to the weight matrix itself. We now present the forward pass for a weight transformation layer layer with $d$ input and $d$ output dimensions and latent dimension $c$. First, we compute $w = W^{\text{wt}}\mathbf{z}$, where $W^{\text{wt}} \in \mathbb{R}^{(d^2+d)\times c}$. The first $d^2$ components of $w$ are reshaped into the $d \times d$ weight matrix of the layer $W^*$, and the last $d$ components are used as the bias vector $b^*$. The forward pass is then the same as a regular fully-connected layer, but using the computed matrix and bias $W^*$ and $b^*$ instead of a fixed matrix and bias vector; that is $y = \sigma(W^*\mathbf{x} + \mathbf{b}^*)$. During adaptation, both the latent vector $\mathbf{z}$ and the transform matrix $W^{\text{wt}}$ are adapted. We note that adapting $\mathbf{z}$ enables the post-adaptation weight matrix used in the forward pass, $W^{*'}$, to differ from the pre-adaptation weight matrix $W^*$ by a matrix of rank up to the dimension of $\mathbf{z}$, whereas gradient descent with normal layers makes rank-1 updates to weight matrices. We hypothesize it is this added expressivity that makes the weight transform layer effective. A comparison of MACAW with and without weight transformation layers can be found in Figures 4-center and 6.

## C  ADDITIONAL EXPERIMENTS AND ABLATIONS

### C.1  WEIGHT TRANSFORM ABLATION STUDY

In addition to the results shown in Figure 4 (center), we include an ablation of the weight transform here for all tasks. Figure 5 shows these results. We find that across environments, the weight transform plays a significant role in increasing training speed, stability, and even final performance. On the relatively simple cheetah direction benchmark, it does not affect the quality of the final meta-trained agent, but it does improve the speed and stability of training. On the other three (more difficult) tasks, we see a much more noticeable affect in terms of both training stability as well as final performance.

Additionally, we investigate the effect of the weight transform in a few-shot image classification setting. We use the 20-way 1-shot Omniglot digit classification setup (Lake et al., 2015), specifically the train/val split used by (Vinyals et al., 2016) as implemented by Deleu et al. (2019). We compare three MLP models, all with 4 hidden layers:

1. An MLP **with weight transform layers** of 128 hidden units and a latent layer dimension of 32 (4,866,048 parameters; Weight Transform in Figure 6).

2. An MLP **without weight transform layers**, with 128 hidden units (152,596 parameters; No WT-Equal Width in Figure 6)

3. An MLP **without weight transform layers**, with 1150 hidden units (4,896,720 parameters; No WT-Equal Params in Figure 6)

We find that the model with weight transform layers shows the best combination of fast convergence and good asymptotic performance compared with baselines with regular fully-connected layers. *No WT-Equal Width* has the same number of hidden units as the weight transform model (128), which means the model has fewer parameters in total (because the weight transform layers include a larger weight matrix). The *No WT-Equal Params* baseline uses wider hidden layers to equalize the number

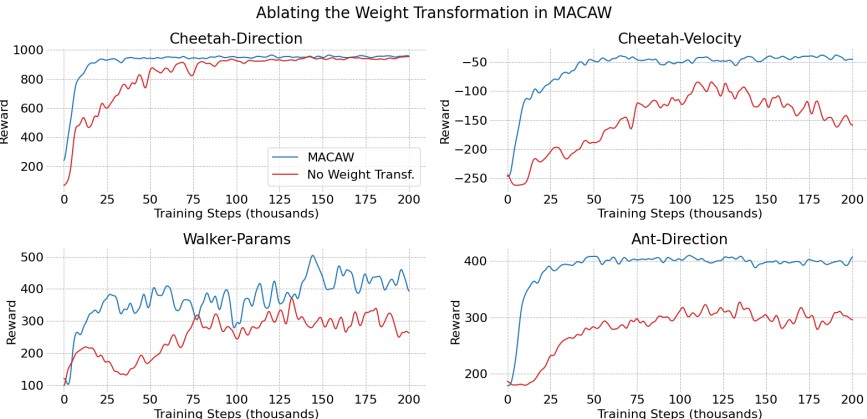

Figure 5: Ablating the weight transformation in MACAW on the MuJoCo benchmark environments. All networks have the same number of hidden units. Although MACAW is able to learn with regular fully-connected layers, the weight transformation significantly improves performance on all tasks that require adaptation to unseen tasks.

of parameters in the entire model with the Weight Transform model. Somewhat surprisingly, the smaller baseline model (Equal Width) outperforms the larger baseline model (Equal Params).

When using MAML-style meta-learners, it is important to consider that adding parameters to the model affects the expressiveness of *both* the forward computation of the model and the updates computable with a finite number of steps of gradient descent.

Generally, increasing the number of parameters in the model should improve the model's ability to fit the training set (because the inner loop of MAML is more expressive), which we observe here. Increasing the expressiveness of the inner loop of MAML can also speed convergence, which we also observe in Figure 6. However, by simply adding neurons to a typical MLP, the post-adaptation model tends to overfit the training set more, as we see in Figure 6. On the other hand, adding parameters through weight transformation layers **increases expressiveness of the adaptation step by enabling weight updates with rank greater than 1 without changing the expressiveness of the forward computation of the model**.

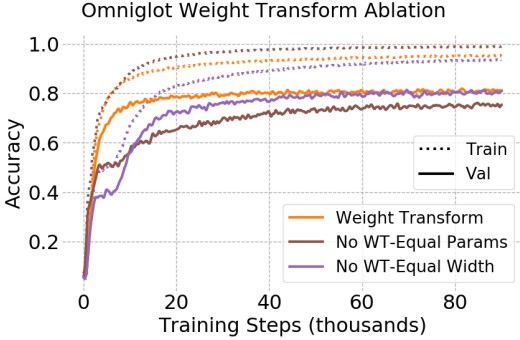

Figure 6: Faster convergence provided by the weight transform layer (orange) on Omniglot 20-way 1-shot image classification (Lake et al., 2015).

### C.2 ONLINE FINE-TUNING FOR OUT-OF-DISTRIBUTION TEST TASKS

In some cases, it may be necessary or desirable to perform **online** fine-tuning after the initial offline adaptation step. This is the fully offline meta-RL problem with online fine-tuning described in Section 3, where an algorithm is given a small amount of initial adaptation data from the test task, just as in the fully offline setting, and then is able to interact with the environment to collect additional training data and perform on-policy updates. This ability to continually improve with additional training after the initial offline adaptation step is what makes a consistent meta-reinforcement learner advantageous.

This hybrid setting (offline training with additional online fine-tuning) is known to be extremely challenging in traditional reinforcement learning. These difficulties are clearly documented by recent work (Nair et al., 2020). In short, this setting is challenging in traditional RL because while offline pre-training might produce a policy that performs well, online fine-tuning often leads to a significant drop in initial performance, which can take a very long time to recover from (see Nair et al. (2020)).

| Additional | Offline PEARL+FT | | MACAW | |
| Env. Steps | Reward | Improvement | Reward | Improvement |
|---|---|---|---|---|
| 0 | -553.4 (21.2) | – | -323.1 (42.9) | – |
| 20k | -565.0 (4.7) | -11.5 (3.5) | -279.1 (16.8) | **44.0 (14.5)** |
| 200k | -533.6 (19.8) | 19.8 (3.7) | -272.0 (15.2) | **51.1 (12.7)** |

Table 1: Absolute reward as well as improvement (in terms of reward) of Offline PEARL+FT and MACAW after 0, 20k, and 200k additional environment steps are gathered and used for online fine tuning. Standard errors of the mean over the 13 test tasks are reported in parentheses. Averages are taken over 10 rollouts of each policy. We find that MACAW achieves both **better out-of-distribution performance** before online training as well as **faster improvement** during online fine-tuning. Note that Offline PEARL+FT experiences an initial *drop* in average performance on the test task after 20k steps, compared with the performance of the policy conditioned only on the initial batch of offline data. A similar effect has been reported in recent work in offline RL (Nair et al., 2020).

In many cases, online fine-tuning can take a very long time to recover the performance of the offline-only policy, if it does so at all. In offline meta-RL, we have a similar challenge; an offline meta-RL algorithm must not only meta-train for good performance on a single batch of offline test data, but it must also learn a set of parameters that enables fine-tuning to make productive updates to its policy and/or value function without completely destroying the meta-learned knowledge about the task distribution.

In this section, we use a hybrid setting as described above to evaluate not only MACAW's consistency (its ability to continue to improve after an initial offline adaptation step), but its ability to continue to improve **even when the test task distribution differs from the train distribution**. Because significant distribution shift means that some train tasks are irrelevant, or even detrimental to test performance, this setting is very difficult. In order to make a meaningful comparison, we compare with an "Offline PEARL + fine-tuning" (Offline PEARL+FT) algorithm, which is also technically consistent (because it essentially performs the SAC algorithm on the test task after the initial task inference step). However, we hypothesize that MACAW will have an advantage over this Offline PEARL+FT algorithm because while both algorithms are consistent, MACAW explicitly trains for good fine-tunability with gradient descent, unlike task inference-based meta-RL algorithms.

The training procedure for Offline PEARL+FT is the same at the regular Offline PEARL training procedure. However, at test time, after receiving an initial small batch of offline data for task inference, we alternative between performing rollouts of the task-conditioned policy to collect additional data from the test task and perform gradient descent on the PEARL policy and value function objectives with this off-policy data. Similarly, for MACAW test time involves first using the small batch of *offline* test task data to take an initial gradient step on the value and policy loss functions (Eqns 2 and 3), then alternating between rolling out the adapted policy and taking more steps of gradient descent on the MACAW losses.

The specific experimental setup is as follows. We partition the individual tasks in the Cheetah-Vel problem such that training tasks correspond to target velocities in the range [0,2] and test tasks correspond to target velocities in the range [2,3]. After meta-training, for each test task, we provide the algorithm with a small batch of offline data for adaptation just as in the fully offline setting. However, we allow the algorithm to then collect and train on up to 200k additional interactions from the environment. Both algorithms alternate between sampling a single trajectory (200 environment interactions) and performing 100 steps of gradient descent on the aggregate buffer of data for the test task, which contains both the initial offline batch of data as well as all online data collected so far. We evaluate both algorithms on their performance after 20k and 200k additional interactions with the environment. The results of this experiment are reported in Table 1. We observe that MACAW achieves both **higher absolute reward** on the OOD test tasks as well as **faster relative improvement** over the offline-only adapted policy compared to the Offline PEARL+FT baseline.

### C.3 METAWORLD ML45 BENCHMARK

As an additional experiment, we test the training and generalization capabilities of MACAW on a much broader distribution of tasks, and where test tasks differ significantly from training tasks (e.g. picking up an object as opposed to opening a window or hammering a nail). Recently,

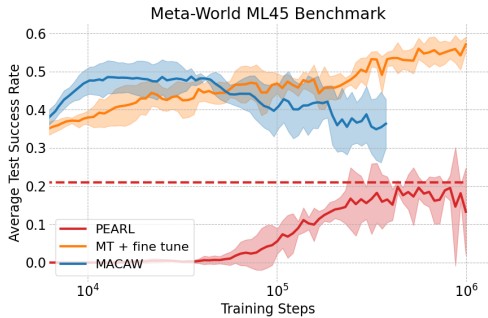

Figure 7: Average success rates of MACAW, PEARL, and MT + fine-tuning (with 20 fine-tuning steps) on the 5 test tasks the Meta-World ML45 suite of continuous control tasks. Dashed line shows final PEARL average success rate after 10m training steps.

Yu et al. (2019) proposed the Meta-World (Yu et al., 2019) suite of continuous control benchmark environments as a more realistic distribution of tasks for multi-task and meta-learning algorithms. This benchmark includes 45 meta-training tasks and 5 meta-testing tasks. The results of this experiment are summarized in Figure 7.

We find that all methods are able to make meaningful progress on the test tasks, with gradient-based methods (MACAW and MT + fine tune) learning much more quickly than PEARL. MACAW does achieve a quite high level of performance quite early on in training; however, it begins to overfit with further training. In the regime where periodic online evaluations are available for the purpose of early stopping, we could avoid this issue, in which case MACAW would slightly underperform the multi-task learning baseline. A possible reason for some inconsistency between the performance of each algorithm on Meta-World and the results reported in Figure 3 is the difficult scaling of the rewards in the current version of the Meta-World benchmark. Rewards can vary by 5 orders of magnitude, from negative values to values on the order of 100,000. This has been documented to adversely impact training performance even in single-task RL and increase hyperparameter sensitivity (see `https://github.com/rlworkgroup/metaworld/issues/226`). Because of the problems stemming from the current reward functions in Meta-World, the maintainers of the benchmark are updating them for the next version of the benchmark, which has not been released as of November 2020.

## D    EXPERIMENTAL SET-UP AND DATA COLLECTION

### D.1    OVERVIEW OF PROBLEM SETTINGS

The problems of interest include:

1. **Half-Cheetah Direction** Train a simple cheetah to run in one of two direction: forward and backward. Thus, there are no held-out test tasks for this problem, making it more 'proof of concept' than benchmark.

2. **Half-Cheetah Velocity** Train a cheetah to run at a desired velocity, which fully parameterizes each task. For our main experiment, values of the task parameters are sampled from a uniform interval of 40 velocities in the range [0, 3]. A subset of 5 target velocities is sampled randomly for evaluation. For ablation experiments

3. **Ant-2D Direction** Train a simulated ant with 8 articulated joints to run in a random 2D direction. For our experiments, we sample 50 random directions uniformly, holding out 5 for testing.

4. **Walker-2D Params** Train a simulated agent to move forward, where different tasks correspond to different randomized dynamics parameters rather than reward functions. For our experiments, we sample 50 random sets of dynamics parameters, holding out 5 for testing.

5. **Meta-World ML45** Train a simulated Sawyer robot to complete 45 different robotics manipulation tasks (for training). 5 additional tasks are included for testing, making 50 tasks in total. Tasks include opening a window, hammering a nail, pulling a lever, picking & placing

| Parameter | Standard Configuration | Meta-World |
|---|---|---|
| Optimizer | Adam | – |
| Meta batch size | 4-10 | 16 |
| Batch size | 256 | – |
| Embedding batch size | 100-256 | 750 |
| KL penalty | 0.1 | – |
| Hidden layers | 3 | – |
| Neurons per hidden layer | 300 | 512 |
| Latent space size | 5 | 8 |
| Policy learning rate | 3e-4 | – |
| Value function learning rate | 3e-4 | – |
| Context embedding learning rate | 3e-4 | – |
| Q-Function learning rate | 3e-4 | – |
| Reward scale | 5.0 | – |
| Recurrent | False | – |

Table 2: Hyperparameters used for the PEARL experiments. For the MuJoCo tasks, we generally used the same parameters as reported in (Rakelly et al., 2019), with some minor modifications. The different parameters used for the MetaWorld ML45 environment are reported above.

> an object. See Yu et al. (2019) for more information. Our experiments use a continuous space randomization for each task setup, unlike the experiments in (Yu et al., 2019), which sample from a fixed number of task states. This creates a much more challenging environment, as seen in the success rate curves above.

For the first 4 MuJoCo domains, each trajectory is 200 time steps (as in Rakelly et al. (2019)); for Meta-World, trajectories are 150 time steps long.

### D.2    DATA COLLECTION

We adapt each task to the offline setting by restricting the data sampling procedure to sample data only from a fixed offline buffer of data. For each task, we train a separate policy from scratch, using Soft Actor-Critic (Haarnoja et al., 2018) for all tasks except Cheetah-Velocity, for which we use TD3 (Fujimoto et al., 2018) as it proved more stable across the various Cheetah-Velocity tasks. We save complete replay buffers from the entire lifetime of training for each task, which includes 5M steps for Meta-World, 2.5M steps for Cheetah-Velocity, 2.5M steps for Cheetah-Dir, 2M steps for Ant-Direction, and 1M steps for Walker-Params. We use these buffers of trajectories, one per task for each problem, to sample data in both the inner and outer loop of the algorithm during training. See Figures 8 and 9 for the learning curves of the offline policies for each train and test task.

### D.3    ABLATION EXPERIMENTS

For the data quality experiment, we compare the post-adaptation performance when MACAW is trained with 3 different sampling regimes for the Cheetah-Vel problem setting. Bad, medium, and good data quality mean that adaptation data (during both training and evaluation) is drawn from the first, middle, and last 500 trajectories from the offline replay buffers. For the task quantity experiment, we order the tasks by the target velocity in ascending order, giving equally spaced tasks with target velocities $g_0 = 0.075$, $g_2 = 0.15$, ..., $g_{39} = 3.0$. For the 20 task experiment, we use $g_i$ with even $i$ for training and odd $i$ for testing. For the 10 task experiment, we move every other train task to the test set (e.g. tasks $i = 2, 6, 10, ...$). For the 5 task experiment, we move every other remaining train task to the test set (e.g. tasks $i = 4, 12, 20, ...$), and for the 3 task experiment, we again move every other task to the test set, so that the train set only contains tasks 0, 16, and 32. Task selection was performed this way to ensure that even in sparse task environments, the train tasks provide coverage of most of the task space.

## E    IMPLEMENTATION DETAILS AND HYPERPARAMETERS

Peng et al. (2019) note several strategies used to increase the stability of their advantage-weighted regression implementation. We normalize the advantage logits in the policy update step to have zero mean and unit standard deviation, as in Peng et al. (2019). Advantage weight logits are also clipped

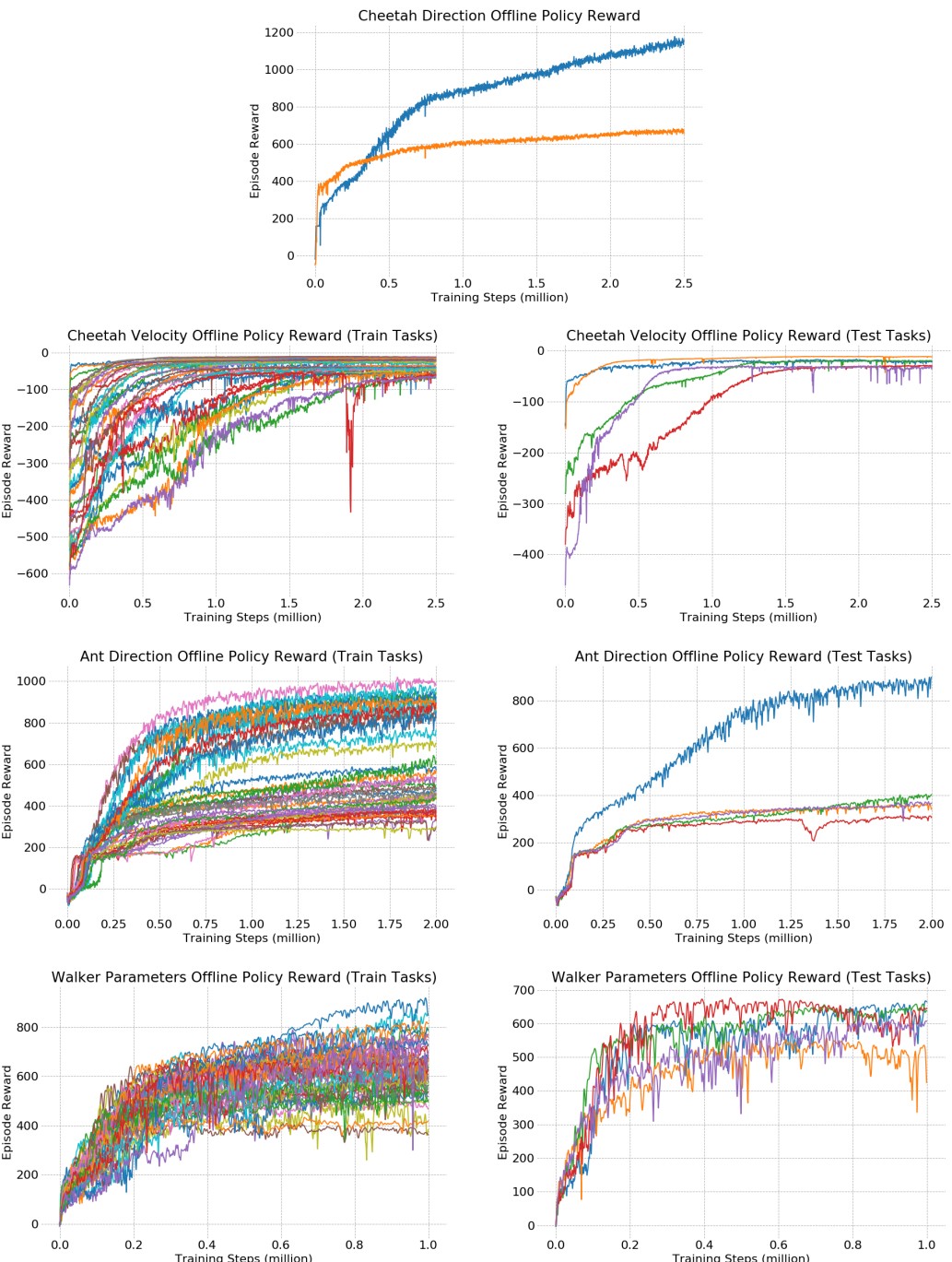

Figure 8: Learning curves for offline policies for the 4 different MuJoCo environments used in the experimental evaluations. Each curve corresponds to a policy trained on a unique task. Various levels of smoothing are applied for the purpose of easier visualization.

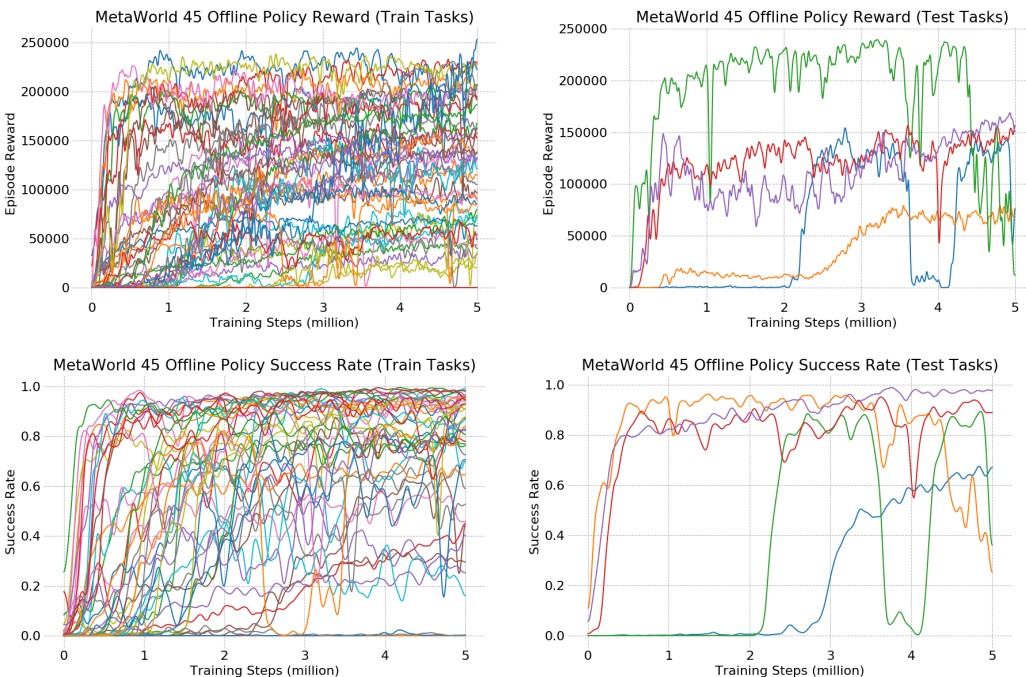

Figure 9: Learning curves and success rates for all tasks in the MetaWorld 45 benchmark. Each curve corresponds to a policy trained on a unique task. Various levels of smoothing are applied for the purpose of plotting.

| Parameter | Standard Configuration | Meta-World |
|---|---|---|
| Optimizer | Adam | – |
| Value learning rate | 1e-4* | 1e-6 |
| Policy learning rate | 1e-4 | – |
| Value fine-tuning learning rate | 1e-4 | 1e-6 |
| Policy fine-tuning learning rate | 1e-3 | – |
| Train outer loop batch size | 256 | – |
| Fine-tuning batch size | 256 | – |
| Number of hidden layers | 3 | – |
| Neurons per hidden layer | 100 | 300 |
| Task batch size | 5 | – |
| Max advantage clip | 20 | – |

Table 3: Hyperparameters used for the multi-task learning + fine tuning baseline. *For the Walker environment, the value learning rate was 1e-5 for stability.

| Parameter | Standard Configuration | Meta-World |
|---|---|---|
| Optimizer | Adam | – |
| Auxiliary advantage loss coefficient | 1e-2 | 1e-3 |
| Outer value learning rate | 1e-5 | 1e-6 |
| Outer policy learning rate | 1e-4 | – |
| Inner policy learning rate | 1e-3 (learned) | 1e-2 (learned) |
| Inner value learning rate | 1e-3 (learned) | 1e-4 (learned) |
| Train outer loop batch size | 256 | – |
| Train adaptation batch size | 256 | 256 |
| Eval adaptation batch size | 256 | – |
| Number of adaptation steps | 1 | – |
| Learning rate for learnable learning rate | 1e-3 | – |
| Number of hidden layers | 3 | – |
| Neurons per hidden layer | 100 | 300 |
| Task batch size | 5 | 10 |
| Max advantage clip | 20 | – |
| AWR policy temperature | 1 | – |

Table 4: Hyperparameters used for MACAW. The Standard Configuration is used for all experiments and all environments except for Meta-World (due to the extreme difference in magnitude of rewards in Meta-World, which has typical rewards 100-1000x larger than in the other tasks). For the Meta-World configuration, only parameters that differ from the standard configuration are listed.

to avoid exploding gradients and numerical overflow. To train the value function, we use simple least squares regression onto Monte Carlo returns, rather than TD($\lambda$). Finally, our policy is parameterized by a single Gaussian with fixed variance of 0.04; our policy network thus predicts only the mean of the Gaussian distribution.

In addition to using weight transformation layers instead of regular fully-connected layers, we also learn learning rates for each layer of our network by gradient descent. To speed up training, we compute our loss using a 'task minibatch' of 5 tasks at each step of optimization, rather than using all of the training tasks. Finally, specific to the RL setting, we sample experiences in contiguous chunks from the replay buffers during train-time adaptation and uniformly (non-contiguously) from the replay buffers for outer-loop updates and test-time adaptation. For outer loop updates, we sample data selectively towards the end of the replay buffers.

E.1 HYPERPARAMETERS

Tables 2, 3, and 4 describe the hyperparameters used for each algorithm in our empirical evaluations. We performed some manual tuning of hyperparameters for all algorithms, but found that the performance was not significantly affected for environments other than Meta-World, likely due to the difficult reward scaling in the current release of Meta-World.

