# OpenReview forum: "Offline Meta-Reinforcement Learning with Advantage Weighting"
_ICLR.cc/2021/Conference — Reject_

### Official Review · AnonReviewer2 · 2020-10-27

**Rating:** 6
**Confidence:** 3

**Review:**

This paper introduces a new problem setting in meta reinforcement learning, namely metaRL. Here the agent is trained on a fixed offline dataset, which distinguishes it from most metaRL algorithms that interact with the environment during meta training. The authors propose a gradient-based meta learning method (MACAW) to approach this problem, which uses an actor-critic method combined with advantage weighting which is an offline RL method. In experiments they show that this outperforms the offline metaRL method PEARL, and combining multi-task offline RL with AWR in a naive way.

Overall I like the paper and think the problem setting is very interesting, and I like the proposed method. I have to admit I was a little bit disappointed when I came to the experiments - they're fine, but not exciting. I also have a few concerns with the practicability of this method when it comes to real-world datasets, which is one of the main motivations for this work. For now I give a score of 5, but I'm open to increase this and look forward to the author's response!

Pros:
- The paper is very well written, and easy to follow. Concepts are well explained and hypotheses clearly stated.
- The proposed problem setting is timely and relevant for the metaRL community.  I like how explicitly the problem setting is described and how it is related to existing work and problem settings.
- The proposed method MACAW is sound and all concepts and objectives explained well. Even though there are quite a few components, I think I would be able to re-implement the method just from the paper description.
- In the experiments on 4 MuJoCo benchmarks the proposed method outperforms two baselines, Offline PEARL and Offline MT+FT.

Cons:
- I feel like the experiment section is not as strong as it could be. All the ingredients are there though and I think with some extra work the authors can easily improve their paper.
 - Throughout the paper, you stress the importance of the algorithm being consistent. However, I feel like there's an experiment missing that shows that MACAW behaves like a consistent algorithm. What I would like to see is, if there is a significant shift between training and task distribution, can MACAW recover from a bad initialisation? How long does it take, and how does this compare to say continuing to train PEARL? Figure 4 (right) slightly hints at this but given that MACAW is still quite good at 3 tasks, and with 3 tasks you can somewhat cover the range of velocities, I think this is not enough. I'm not even entirely convinced by this "consistency" argument (especially not with empirical evidence). The agent only has a small dataset available at test time, so why care about consistency? Consistency is *only* useful if I can actually train longer than that - but in this case, I can also just continue training whatever other metaRL algorithm I pre-trained (like PEARL, etc). So what does consistency buy me?
 - The paper could also be considerably improved with a more realistic benchmark. A main motivation for the offline RL setting is that you can use real-world data and train a metaRL agent using this. So why not use something more realistic (even something like meta-world would already be cool)? Or maybe there exists an off-line dataset with real-world data where you don't actually execute the policy in the end (because you can't) but you can somehow compute how well it does (by comparing it with expert actions for example).
- I wonder if considering the *fully* offline dataset makes the problem too easy. I can imagine that for many real-world applications where you want to deploy the agent on new tasks, it will have to gather the data to learn about the task by itself, instead of being given this in an offline fashion (since that might often not be possible!). The authors briefly discuss this at the end of the conclusion and cite concurrent work by Dorfman and Tamar (2020) who aim to do this. Would it be possible to extend MACAW to learn good exploration policies (doesn't even have to be online, there could be a separate exploration and exploitation phase)?

Other questions:
- Are the authors concerned that the updates for the value function (Eq 2) suffers from high variance due to the Monte Carlo returns? In the MuJoCo tasks considered in the experiments this might not be such a problem due to (a) dense rewards and (b) "good" offline data. But could this become a problem if those two things are not given? How would you deal with this?
 - Given that the training data is pretty good since it comes from policies that are pre-trained (using "good data"), would it make sense to add a simpler, supervised imitation learning algorithm as a baseline? Since we're in the *fully* offline metaRL setting this should be possible?
- Would MACAW also work on the standard metaRL setting, where the agent is interacting with the environment during meta training and therefore responsible for collecting the data itself?

Side comments:
- Sec 5: I don't understand the second sentence; do you mean "on" this problem? Why is MACAW listed as a "sensible" approach?
- Fig 4 left: The colours in the caption don't match the one in the figure.

-------------------------------------------------------------------------------------------------------------------------
UPDATE

I have read the other reviews and the author's response.

Thank you for your thorough answer! It's great to see you've taken all feedback into account and updated the paper significantly. After looking through the changes in the paper I'm raising my score from 5 to 6.

Some last comments:
- Several other reviewers also raised concerns that the "fully" offline setting might be unrealistic. I saw you added a motivation for this in Sec 3, which makes me a little bit more convinced. It would still be great to add an experiment where MACAW is adapted online at test time (entirely without offline data) like in C.2 but on in-distribution tasks.
- I understand how the reward function issues in ML45 could be the cause of the inconsistent results. Maybe ML1 would be a better choice at this point, or indeed waiting until v2 of the benchmark is released.

---

> ### Author Response · Authors · 2020-11-19
> **Response to Reviewer 2**
>
> Thank you for your very useful feedback! We've revised the paper and believe that we've addressed your concerns. If you have any remaining questions, please let us know.
>
> "I feel like the experiment section is not as strong as it could be. All the ingredients are there though and I think with some extra work the authors can easily improve their paper."
>
> We have performed several additional experiments:
> A meta-imitation baseline to our main results (Figure 3), showing that MACAW is the only algorithm to consistently improve upon imitation learning
> An additional ablation of the weight transform in both a meta-RL as well as a supervised learning setting, to more clearly show its utility in boosting the speed of convergence of MAML (Figure 4-center and Appendix C.1 Figure 6)
> An experiment comparing the out of distribution performance of PEARL and MACAW as well as their performance when allowed some online fine-tuning, showing that MACAW generalizes better to OOD test tasks and that MACAW takes advantage of online fine-tuning after the initial offline adaptation stage better than PEARL (Appendix C.2 Table 1)
>
> "...can MACAW recover from a bad initialisation?...I can also just continue training whatever other metaRL algorithm I pre-trained (like PEARL, etc). So what does consistency buy me?"
>
> See the new OOD generalization/online fine-tuning experiment in Appendix C.2 Table 1. We find that MACAW handles distribution shift and fine-tunes faster than Offline PEARL.
>
> "The paper could also be considerably improved with a more realistic benchmark"
>
> Figure 7 in Appendix C.3 shows the results of an experiment on Meta-World. The results in this setting are somewhat mixed, and as we have noted to other reviewers, we believe this is due to problematic reward functions in the Meta-World benchmark. This recent github issue identifies exactly this problem: https://github.com/rlworkgroup/metaworld/issues/226 (note: the person who raised the issue is not affiliated with our paper). A maintainer of the project has confirmed this issue and stated that the rewards are currently being rewritten for this reason, and that an updated benchmark should be released soon. Thus we believe that the poor scaling of rewards makes our Meta-World results extremely sensitive to hyperparameters; coupled with the large number of hyperparameters that meta-RL algorithms tend to have (even compared to RL), this likely explains the inconsistency in this result compared with our other experiments. We have explained this issue in our Appendix to better contextualize our Meta-World results.
>
> "Would it be possible to extend MACAW to learn good exploration policies (doesn't even have to be online, there could be a separate exploration and exploitation phase)?"
>
> The algorithm described by Dorfman and Tamar (2020) leverages the assumption that the offline data can be relabeled with the rewards from all other tasks, which greatly broadens the data distribution available for meta-training, which we do not assume in MACAW. Under this assumption, it is possible that MACAW could learn an importance weighting-based exploration policy (utilizing the much broader training distribution). In general, learning online exploration in the offline setting is very difficult, and investigating this sort of extension to MACAW is a very interesting direction for future work. We have updated the conclusion to point this out more clearly.
>
> "In the MuJoCo tasks considered in the experiments this might not be such a problem due to (a) dense rewards and (b) "good" offline data. But could this become a problem if those two things are not given?"
>
> High variance updates from sub-optimal data can in fact reduce the effectiveness of training. However, as we show in Figure 4-left, while sub-optimal data does reduce MACAW's performance, MACAW is still able to learn a reasonable policy (which performs better than PEARL, even when PEARL is provided with higher quality data). To handle trajectories with very little reward signal, we could replace the MAML subroutine of MACAW with an algorithm such as latent embedding optimization to provide a better value function initialization for adaptation.
>
> "Would it make sense to add a simpler, supervised imitation learning algorithm as a baseline?"
>
> Thank you for this suggestion- see the updated Figure 3. We find that MACAW outperforms imitation learning on all tasks.
>
> "Would MACAW also work on the standard metaRL setting?"
>
> Yes; MACAW is not intrinsically limited to the offline setting. We have emphasized the offline setting due to its practical significance and relative lack of attention in the community.
>
> We hope these comments have addressed your concerns- if not, please let us know!
>
> [1] Scott Fujimoto, David Meger, Doina Precup. Off-Policy Deep Reinforcement Learning without Exploration
> [2] Aviral Kumar, Justin Fu, George Tucker, Sergey Levine. Stabilizing Off-Policy Q-Learning via Bootstrapping Error Reduction

---

### Official Review · AnonReviewer4 · 2020-10-28

**Rating:** 6
**Confidence:** 3

**Review:**

The paper proposes the problem of fully offline meta-RL. Here, the idea is to leverage offline experience from multiple tasks to enable fast adaptation to new tasks. The paper distinguishes two settings of offline meta-RL, one where only the training data is collected offline and testing corresponds to sampling online trajectories, the other where both training and testing data are collected offline. The latter is termed as fully offline meta-RL and is the problem setting considered in this work.
The paper also proposes a method for the fully offline meta-RL problem based on the MAML method. They argue that a naive application of MAML with advantage weighted regression (a recently proposed approach to offline RL) is insufficient for this setting and propose to make the policy more expressive by including an advantage head that regresses the advantage conditioned on the state and action. The approach is evaluated on offline versions of 4 continuous control problems.

Strong Points
- The paper is well written and the problem setting is well explained.
- A solution is proposed for the fully offline meta-RL problem. The modifications to the policy functions are backed by theory and is also empirically verified to be helpful in the experiments.
- Extensive ablations on the various modifications to MAML+AWR confirm that the utility of the approach for the fully offline meta-RL problem.
- The authors also explore settings of good/bad adaptation data showing the robustness of the proposed method to quality of offline adaptation data as compared with MAML+AWR

Weak Points
- While the offline meta-RL problem setting is well motivated, I am not certain that the “fully offline” setting is as interesting. Since the idea behind meta-RL methods is to adapt to new tasks quickly, it should be generally feasible to have a small number of online trajectories for adaptation.
- The benchmarks used for evaluation are really toy benchmarks. I would have liked to see performance on MetaWorld benchmark. Interestingly, this evaluation is present in the Appendix where the approach seems to perform worse. Moreover, one of the continuous control benchmarks (cheetah direction) in the main paper doesn’t have a held-out test and is more a proof of concept.

Overall, I find the ideas proposed quite interesting. But given my hesitation with the utility of the fully offline setting as well as the experiments, I am recommending a weak accept.

Questions for the authors:
- Can you elaborate on why PEARL doesn’t work at all on even the Cheetah-direction evaluation which doesn’t have any held-out test set?
- How many step of gradient descent are used in the inner loop for the MACAW method? How many steps of fine-tuning for the MT+FT baseline?
- Why is the MetaWorld evaluation not part of the main paper? Can you please comment on the worse performance of MACAW on this more realistic benchmark for meta-RL?
- It will also help if the authors can provide more motivation for the "fully" offline meta-RL setting.

---

> ### Author Response · Authors · 2020-11-19
> **Response to Reviewer 4**
>
> Thank you for your feedback. We believe that we have addressed all of your concerns below, but please let us know if this is not the case.
>
> "While the offline meta-RL problem setting is well motivated, I am not certain that the “fully offline” setting is as interesting. Since the idea behind meta-RL methods is to adapt to new tasks quickly, it should be generally feasible to have a small number of online trajectories for adaptation...It will also help if the authors can provide more motivation for the "fully" offline meta-RL setting."
>
> We have performed additional experiments showing that MACAW performs well in the online fine-tuning regime as well as the fully offline setting. Table 1 in Appendix C.2 shows these results, in which MACAW both a) generalizes better to out of distribution test tasks and b) improves more quickly when training on additional *online* data after the initial offline adaptation stage. We have also revised Section 3 to consider the possibility that, after the initial offline adaptation phase, the adapted policy might adapt further using the online data gathered from deployment. In terms of motivating the fully offline setting, we note that in safety-critical settings, allowing a meta-trained policy to do its own initial exploration in a test environment might not be possible or safe. In these cases, it might be more reasonable for an expert or carefully crafted heuristic to produce a small amount of offline test data for initial adaptation. However, after this initial offline adaptation step, it is possible that further updates with online data might prove beneficial. Our additional experiment in Appendix C.2 is aimed at this setting.
>
> "The benchmarks used for evaluation are really toy benchmarks. I would have liked to see performance on MetaWorld benchmark. Interestingly, this evaluation is present in the Appendix where the approach seems to perform worse. Moreover, one of the continuous control benchmarks (cheetah direction) in the main paper doesn’t have a held-out test and is more a proof of concept."
>
> Regarding Meta-World:
>
> We believe this result is due to problematic reward functions in the Meta-World benchmark. A recent github issue on the metaworld project (which we became aware of after submitting our paper) discussing the poor conditioning of the rewards in many metaworld environments can be found here: https://github.com/rlworkgroup/metaworld/issues/226 (note: the person who raised the issue is not affiliated with our paper). The user raising the issue points out that the reward scaling makes learning very difficult even in the single-task setting, and that the results become very sensitive to hyperparameters. A maintainer of the project has confirmed this issue and stated that the rewards are currently being rewritten for this reason, and that an updated benchmark should be released soon. Thus we believe that the poor scaling of rewards makes our Meta-World results extremely sensitive to hyperparameters; coupled with the large number of hyperparameters that meta-RL algorithms tend to have (even compared to RL), this likely explains the inconsistency in this result compared with our other experiments. We have explained this issue in our Appendix to better contextualize our Meta-World results.
>
> Regarding Cheetah-Dir:
>
> We agree that Cheetah-Dir is quite a simple environment. However, we feel it is worth reporting the results on this environment in the offline setting, because even on a task as easy as Cheetah-Dir, we find that there is a significant difference between the performance MACAW and the prior methods.
>
> "Can you elaborate on why PEARL doesn’t work at all on even the Cheetah-direction evaluation which doesn’t have any held-out test set?"
>
> Algorithms that perform well in the online or off-policy RL setting can fail in the offline setting (see [1], [2], [3], [4] for more discussion about this problem).
>
> "How many step of gradient descent are used in the inner loop for the MACAW method? How many steps of fine-tuning for the MT+FT baseline?"
>
> MACAW uses a single gradient step for adaptation. The MT+FT baseline uses 20 steps of Adam for adaptation. This is stated in Section 5, but we have clarified this in the captions of Figure 3 as well.
>
> We hope these clarifications have addressed your questions. If not, we look forward to discussing them with you further.
>
> [1] Scott Fujimoto, David Meger, Doina Precup. Off-Policy Deep Reinforcement Learning without Exploration
> [2] Aviral Kumar, Aurick Zhou, George Tucker, Sergey Levine. Conservative Q-Learning for Offline Reinforcement Learning
> [3] Chelsea Finn & Sergey Levine. Meta-Learning and Universality: Deep Representations and Gradient Descent can Approximate any Learning Algorithm
> [4] Rahul Kidambi, Aravind Rajeswaran, Praneeth Netrapalli, Thorsten Joachims. MOReL: Model-based Offline Reinforcement Learning

---

### Official Review · AnonReviewer3 · 2020-10-28
**Well-written paper on a nice topic, concerns about formalization and weight transform**

**Rating:** 5
**Confidence:** 4

**Review:**

- Summary:
    - This paper makes two contributions:
        - 1. Formalizing the offline meta-RL paradigm, where we meta-train on pre-collected (offline) data for several RL tasks and adapt to a new task with a small amount of data. Within offline meta-RL the experiments focus on the fully offline case, where the meta-test task is also offline. It could be online with the meta-train tasks remaining offline, but then we would have to meta-learn an exploration policy, which isn't done in this work.
        - 2. Introducing MACAW: an algorithm for offline meta-RL that has the desirable property of being consistent (i.e. converges to a good policy if enough time and data for the meta-test task are given, regardless of meta-training). To do so they rely on MAML (which provides consistency) and AWR (a simple, popular offline RL algorithm) and add a couple of changes: some hyper-network like parameterization to add capacity and adding an extra objective in the policy update to enrich the inner loop.

- Pros:
    - 1. The paper proposes an important area of research
    - 2. Most of the experiments are well executed, using good baselines and as well as providing understanding through ablations
    - 3. MACAW is a nice simple algorithm with good guarantees.

- Cons:
    1. I think the offline meta-RL paradigm is not introduced correctly.
        - In particular, the paper largely borrows the meta-RL formulation from the online setting where task=MDP. It then treats the collection of the batch data as an obvious after-thought once the task is defined. However, in the offline setting the policy that generated the batch of data is of critical importance and should be part of the task definition.
        - For instance, it is not the same to receive an MDP and examples from the perfect policy (where you can take supervised examples of a given s) than from a random policy (where there is no signal) or an adversarial policy that tries to act as bad as possible. This also has consequences at the meta-level: if at meta-training time I only see examples of perfect policies I may learn to imitate them, then at meta-test time I see an in-distribution MDP but data coming a bad policy and imitating it is a bad idea. In the definition of 'task' described in the paper this is fine since task=MDP. However, we would expect this to work poorly since the policy is out-of-distribution with those since at meta-training.
        - This effect of the quality of the data is used in the experiment of figure 4 left, which makes the experiment good (a Pro), but does not detract from having (IMO) the wrong formalization.
        - This is a big minus since this formalization is one of the big contributions of the paper and it could affect further papers in that area. However, there is a chance I am wrong since the authors have spent months on this and I've spent only some hours reviewing the paper.

    2. One of the two improvements over MAML+AWR, the weight transform, is not fully justified:
        - From a conceptual point of view, it's not clear to me why this is the first time MAML has needed this change after being used in tens/hundreds of experiments. What is different on this task that hasn't been true in any other task in the past? I understood we're doing it to increase the representation capabilities of the gradient, but wouldn't this be useful for other meta-learning tasks? If so, why wasn't it used in the past? (specially since the bias-only version of the idea was already proposed by Finn et al. in 2017)
        - From an experimental point of view I couldn't find the details of the ablated version on the main text appendix B, D or E. Therefore, I understood we're simply changing it to just a weight matrix of the same dimension. If that's the case one could argue that to make things comparable we should try increasing the width by a factor of $\sqrt{c}$ (c defined as in the appendix) to have roughly the same number of parameters, as well as possibly the depth, since that would be a simpler change with a similar latency to the weight transform version.

    3. [edited post-rebuttal to correct inaccuracy] The fact that there is a recent/concurrent paper is not ideal. However, I didn't weight it in my consideration.
- Clarity: pretty high
- Significance: somewhat high, except for similar concurrent NeurIPS work
- Questions:
    - Figure 4 left I didn't understand if the quality changed at meta-test only or both at meta-train and meta-test.
    - Any conjecture on why PEARL performance goes down in cheetah-velocity?
    - Doesn't AWR rely on the policy providing the data being somewhat good already? Otherwise the advantage function may be very different for the policy that generated the data vs. the optimal policy.
- Details:
    - Figure 3 has logarithmic x axis and figure 4 (left, center) have linear x axis. It may be better to keep them the same.
    - "An important property of a meta-RL algorithms is thus its robustness" --> _offline_ meta-RL?
    - In Related work Kirsch 2020b,a seems to be the same paper cited twice
- Summary of review: because I believe the formalization has a big flaw and this paper is mainly about the formalization, I have to recommend rejection. I also have major concerns regarding the weight transform and the experiments that were done to prove its usefulness. The paper is otherwise good, interesting, well-written, and timely; I'm looking forward to the discussion and updating my review if my initial assessment was wrong.

============
Update after discussion with authors

I had two main concerns:
- The modification to MAML was unconvincing to me.
- The offline meta-RL formulation should include behavior policy as part of the task definition.

After a very detailed response from the authors, I am now happy with the response and extra experiments w.r.t. the MAML modification, but I still have concerns about the formulation. In particular, reading the final version I still think the policy giving the behavior data is treated as an after-thought and is instead assumed constant across all tasks. For instance, IMO figure 1 should contain multiple examples of the same "RL task" that are different "offline RL tasks"; i.e. learning to swim using guidance from a 3-year-old and learning to swim using guidance from Michael Phelps. This is one of the key differences, IMO, between meta-RL and offline meta-RL and given that this paper's main contribution is introducing offline meta-RL, I feel it really should be very clear about this point. It may be fine to first introduce the correct general version and then say something like "it may be useful to assume each RL task is given by an expert of roughly the same characteristics", i.e. we can assume behavior policy is constant across tasks. However, right now the original formulation directly borrows from regular meta-RL and I believe that may confuse future papers in offline meta-RL.

I've increased my score from 4 to 5 since I'm now less concerned about the MAML improvement, but I cannot recommend acceptance given my concern about the formulation.

---

> ### Author Response · Authors · 2020-11-19
> **Response to Reviewer 3**
>
> Thank you for your very thoughtful response. We believe that we have addressed all of your concerns below, but please let us know if this is not the case.
>
> "However, in the offline setting the policy that generated the batch of data is of critical importance and should be part of the task definition."
>
> We agree that the distribution of offline data available at meta-train and meta-test time is very important (this motivated the data quality ablation in Figure 4-left, as you pointed out). We have clarified the problem statement in Section 3 to explicitly link the offline data buffers B_i to behavior policies, to emphasize that while the objective in any case is the same (maximize returns), the maximum achievable test returns will differ for different offline data distributions.
>
> "From a conceptual point of view, it's not clear to me why this is the first time MAML has needed this change after being used in tens/hundreds of experiments."
>
> We have conducted additional experiments to show that the weight transform is a generally beneficial augmentation to MAML, outside of the context of meta-RL, finding that it significantly speeds convergence in supervised settings as well (when controlling for total number of parameters, as you mentioned). See the updated Figure 4-center (which now contains an "equal params" baseline) and Figure 6 in Appendix C for meta-RL and supervised learning results, respectively. Building from the analysis of [3], we hypothesize that the low-rank updates that typical MAML updates make can be insufficient in some cases (because a single gradient step can make only a rank-one update to each weight matrix). [3] considers architectures that have depth larger than width for this reason, which is not practical; the weight transform enables us to increase the rank of the updates to our network at each gradient step without affecting the capacity of the function that the network represents. We have added more discussion about this rank limitation in Appendix B, where the weight transform is described.
>
> "Figure 4 left I didn't understand if the quality changed at meta-test only or both at meta-train and meta-test."
>
> The data quality has changed at both meta-train and meta-test time. Thanks for pointing this out- we have clarified it in both the caption of Figure 4 and the text of Section 5 in the revision.
> "Any conjecture on why PEARL performance goes down in cheetah-velocity?"
>
> This type of behavior (where the policy initially improves but then significantly deteriorates) has been observed in the offline RL literature, and we assume a similar issue is occurring here (see [1], [2], [3] for more discussion about this problem).
>
> "Doesn't AWR rely on the policy providing the data being somewhat good already? Otherwise the advantage function may be very different for the policy that generated the data vs. the optimal policy."
>
> You're correct to point out that any offline algorithm will be impacted by the quality of the offline data, but AWR is generally able to outperform the behavior policy, sometimes significantly (see Figure 7 in the AWR paper [4]). In our data quality ablation in Figure 4-left, we find that even when inner loop adaptation data is poor, MACAW can still provide pretty good performance on the test tasks.
>
> "The fact that there is a recent/concurrent NeurIPS 2020 paper is not ideal."
>
> We assume that you are referring to the concurrent work by Dorfman & Tamar (if you are referring to a different paper, please let us know!). This paper is not appearing at NeurIPS 2020, to our knowledge.
>
> We have also made several changes to the paper in response to your remarks about style, which are much appreciated.
>
> We hope these clarifications have addressed your concerns. If not, we look forward to discussing them with you further.
>
> [1] Scott Fujimoto, David Meger, Doina Precup. Off-Policy Deep Reinforcement Learning without Exploration
> [2] Aviral Kumar, Aurick Zhou, George Tucker, Sergey Levine. Conservative Q-Learning for Offline Reinforcement Learning
> [3] Chelsea Finn & Sergey Levine. Meta-Learning and Universality: Deep Representations and Gradient Descent can Approximate any Learning Algorithm
> [4] Xue Bin Peng, Aviral Kumar, Grace Zhang, Sergey Levine. Advantage-Weighted Regression: Simple and Scalable Off-Policy Reinforcement Learning

---

### Official Review · AnonReviewer1 · 2020-11-01
**My review**

**Rating:** 5
**Confidence:** 5

**Review:**

This paper proposes a method for "fully" offline meta-RL. Specifically, they assume there is no interaction with the environment at all neither during meta-train nor meta-test and this method only sees previously collected data at all times. Their method is built on top of AWR [1] in which policy updates are weighted by the advantage term.  Except for section 4.3 and the last paragraph of section 4.1 (see below), the paper is well-written and easy to follow.

My comments:
- While this paper touches on a very interesting and practical problem in meta-rl and batch-rl, I didn't find their setting is very realistic with respect to batch and offline RL setup. In batch RL, even though we assume there is a fixed data for training ( no interaction with an environment whatsoever), the final performance will be measured by interaction with the environment (i.e. policy will be evaluated in an online setting). However, this paper assumes that there is no such interaction with the environment exists. The reason the original setup of batch-rl is important because it imposes lots of challenges such as extrapolation error, over-estimation, etc which are very critical in real-worlds and make batch-lr an important problem. However, in this paper, this is not the case. The question is how the setup in this paper is sensible and important at all?

- I am not sure about the "consistent" definition in this work and it doesn't make sense to me. Per paper definition, if an algorithm can find a good solution to test tasks regardless of the meta-training task distribution, is called consistent. My understanding from this definition is training tasks are not important at all and test-tasks can have a very different distribution from training data and still work. If that is true, why we need training tasks in the first place? based on this definition, can't we just randomly initialize the algorithm and do a meta-test and get the same results? (it would be a good experiment to do as well)

- For the experiment section, this paper uses the benchmark introduced by [2, 3] and it compares with PEARL. Did you follow PEARL setup for the experiments or ProMP/MAML?  PEARL samples a fixed set of tasks at the beginning of training while ProMP and MAML samples a set of tasks at every iteration. It is important because PEARL works in the former setups not the latter. In addition, you used the same parameters as PEARL for the experiments in this paper which is not fair as experiments (i.e. fixed dataset) in this paper are different from PEARL. PEARL might get better result with hyper-parameter tuning.

- I am not convinced that why one should select L_\pi in the inner loop instead of L_{AWR}? why not just use L_\pi in both the inner and outer loop? Can you explain the motivation for this choice? Section 4.1 didn't make the case for this selection.

- Experiments are done for only 1M steps. Why 1M for all algorithms as they are different? it is like supervised learning,  you can train for more gradient setups and it is likely results will change ( with hyper-paramaters tuning).

- Looking at Figure 6, MACAW doesn't show a good performance vs. others. Why this is the case? is there anything special about this experiment that MACAW doesn't work?

- What is the difference between R(s,a) in eq.1 and R(s) eq 2? shouldn't  R(s) in eq 2 be R(s,a)?

minor: Figure 7 and 8 don't have legends.

[1] Advantage-weighted regression: Simple and scalable off-policy reinforcement learning, 2019

[2] Chelsea Finn, Pieter Abbeel, and Sergey Levine. Model-agnostic meta-learning for fast adaptation of deep networks

[3] Jonas Rothfuss, Dennis Lee, Ignasi Clavera, Tamim Asfour, and Pieter Abbeel. Promp: Proximal meta-policy search.

---

> ### Author Response · Authors · 2020-11-19
> **Response to Reviewer 1**
>
> Thank you for the feedback! We believe that we have addressed all of your comments below, but please let us know if this is not the case.
>
> "In batch RL...the final performance will be measured by interaction with the environment...However, this paper assumes that there is no such interaction with the environment exists."
>
> We do measure final performance by evaluating each policy in the online setting, as you described. Specifically, only offline data is used for adaptation, while we use online samples to evaluate the adapted policy for each method (this is the reward values shown in Figure 3). We have revised Section 3 to clarify this point.
>
> "...why we need training tasks in the first place? based on this definition, can't we just randomly initialize the algorithm and do a meta-test and get the same results?"
>
> We would like an algorithm capable of both (1) few-shot adaptation on in-distribution tasks and of (2) improvement with more data when faced with out-of-distribution tasks. Random initialization + RL training achieves only (2), without achieving (1). We have updated Section 1 to clarify this. See Table 1 in Appendix Section C.2 for a new experiment with OOD test tasks, in which MACAW is able to adapt and learn online much more quickly than Offline PEARL combined with online data collection and fine tuning.
>
> "Did you follow PEARL setup for the experiments or ProMP/MAML?"
>
> We follow the PEARL setup. Details are provided in Appendix Section C.1. We have revised the paper to clarify this point in the main text as well.
>
> "In addition, you used the same parameters as PEARL for the experiments in this paper which is not fair as experiments (i.e. fixed dataset) in this paper are different from PEARL."
>
> We performed manual hyperparameter tuning for both MACAW and PEARL. The only environment in which we found hyperparameters to make a significant difference was Meta-World (see our later comment about poor reward scaling and hyperparameter sensitivity). Because of the sheer number of significant hyperparameters meta-RL algorithms tend to have, and the computational requirements of each training run, a truly comprehensive hyperparameter sweep was beyond our available compute. We have clarified this point in our revision in the Implementation Details in Appendix E.
>
> "I am not convinced that why one should select $L_\pi$ in the inner loop instead of $L_{AWR}$...Can you explain the motivation for this choice? Section 4.1 didn't make the case for this selection."
>
> In Section 4.1, we state that "The gradient of the AWR objective does not contain full information of both the regression weight and the regression target...To address this issue and make our meta-learner sufficiently expressive, the MACAW policy update performs both advantage-weighted regression onto actions as well as an additional regression onto action advantages." A thorough theoretical analysis of the expressiveness problem with the AWR policy loss as well as the proof that MACAW's proposed enriched policy loss addresses this problem is provided in Appendix A.  We have updated the text in Section 4.1 to clarify this.
>
> "Experiments are done for only 1M steps."
>
> Due to the computational demands of running each algorithm on each environment with 4 seeds, running for significantly longer than 1 million steps was not feasible. Compared to the online setting, it is reasonable to expect faster learning in the offline setting because the data distribution is static, and the agent does not need to do exploration (for example, see the BCQ offline RL experiments in Fujimoto et al., 2018).
>
> "Looking at Figure 6, MACAW doesn't show a good performance vs. others. Why this is the case? is there anything special about this experiment that MACAW doesn't work?"
>
> We believe this result is due to problematic reward functions in the Meta-World benchmark. A recent github issue on the metaworld project discussing the poor conditioning of the rewards in many metaworld environments can be found here: https://github.com/rlworkgroup/metaworld/issues/226 (note: the person who raised the issue is not affiliated with our paper). The user raising the issue points out that the reward scaling makes learning very difficult even in the single-task setting, and that the results become very sensitive to hyperparameters. A maintainer of the project has confirmed this issue and stated that the rewards are currently being rewritten for this reason, and that an updated benchmark should be released soon. Thus we believe that the poor scaling of rewards makes our Meta-World results extremely sensitive to hyperparameters; coupled with the large number of hyperparameters that meta-RL algorithms tend to have (even compared to RL), this likely explains the inconsistency in this result compared with our other experiments. We have explained this issue in our Appendix to better contextualize our Meta-World results.
>
> We have also corrected the typographical errors you noted.

---

### Author Response · Authors · 2020-11-19
**Overview of Revisions**

Thank you to the reviewers for their useful suggestions! We've made some significant improvements to the paper, including:
- Updated Appendix C.2: A new experiment comparing MACAW and Offline PEARL with online fine-tuning on out-of-distribution test tasks. We find that MACAW both a) generalizes better to out of distribution tasks and b) improves more rapidly than PEARL when both algorithms are allowed to gather online data for fine-tuning
- Updated Figure 4-center and Figure 6 in Appendix C.1: New weight transform ablations. Figure 6 shows an ablation of the weight transform in a supervised setting, showing that the weight transform significantly speeds up convergence in this setting as well (controlling for total number of parameters).
- Updated Figure 3: A new meta-imitation baseline for the main experiments. This result shows that MACAW is the only algorithm that consistently outperforms meta-behavior cloning
- Updated Figure 7 (Meta-World experiment in Appendix C.3) with multiple random seeds and to use the same fine-tuning procedure for both multi-task learning + fine-tuning and for MACAW. MACAW performs more competitively in this comparison, though we note some general problems with the implementation of the Meta-World environments that make interpreting the result of this experiment difficult.

In the text, Appendix C has been substantially extended with new experiments and discussion. We have also made some clarifications in the text regarding the relationship between offline data buffers and the behavior policies that generated them (Section 3), the desirability of consistency in (Section 1), the motivation for the MACAW policy loss (Section 4.1), and contextualizing the results of our Meta-World experiment (Appendix C.3). We have also made various fixes to the writing to improve general readability. Finally, we fixed a minor bug in our data sampling procedure for MACAW and some baselines and updated Figures 3 and 4-right accordingly; the results of these experiments (that MACAW outperforms all baselines in Figure 3 and that MACAW is the only algorithm to learn useful priors with extremely sparse task sampling) are unchanged.

We look forward to hearing any additional thoughts or feedback that the reviewers might have on the updated paper.

---

### Decision · Program_Chairs · 2021-01-07
**Final Decision**

**Decision:**

Reject

**Comment:**

The paper proposes a method for offline meta-RL, where we meta-train on pre-collected offline data for several RL tasks and adapt to a new task with a small amount of data. The paper assumes that there is no interaction with the environment either during meta-train or meta-test.  In this setting, motivated by the ide of leveraging offline experience from multiple tasks to enable fast adaptation to new tasks, the paper introduces MACAW, which combines the consistent MAML and the popular offline AWR, improving upon them by adding capacity through parameterization and adding an extra objective in the policy update. As a result, the MACAW proposed for the offline meta-RL has the desirable property of being consistent, i.e., converging to a good policy if enough time and data for the meta-test task are given, regardless of meta-training.


Pros:
+ Most of the experiments are well executed, using good baselines. Extensive ablations on the various modifications to MAML+AWR confirmed the utility of the approach for the fully offline meta-RL problem.
+ MACAW is a simple algorithm with theoretical guarantees; the modifications to the policy functions are backed by theory.


Cons:
- The reviewers have concerns on the formulation of offline meta-RL. One major contribution of the paper is to introduce offline meta-RL. However the paper largely borrows the meta-RL formulation from the online setting where task=MDP. The reviewers think that directly borrowing from regular meta-RL as the formulation of offline meta-RL might be misleading. The reviewers suggest including behavior policy as part of the task definition for offline meta-RL formulation.

- Several reviewers raised concerns that the fully offline setting might be unrealistic. Although the author did add a motivation, the reviewers would be interested in seeing MACAW being adapted online at test time on in-distribution tasks.

- Unfortunately, the authors accidentally revealed their names in one of the modified versions.